# MeteoRA: Multiple-tasks Embedded LoRA for Large Language Models

**Jingwei Xu**[1*]  **Junyu Lai**[1]    **Yunpeng Huang**[1]
[1]State Key Laboratory for Novel Software Technology, Nanjing University, China
`jingweix@nju.edu.cn`, `{junyu_lai,hyp}@smail.nju.edu.cn`

## Abstract

The *pretrain+fine-tune* paradigm is foundational for deploying large language models (LLMs) across various downstream applications. Within this framework, Low-Rank Adaptation (LoRA) stands out for its parameter-efficient fine-tuning (PEFT), producing numerous reusable task-specific LoRA adapters. However, this approach requires explicit task intention selection, posing challenges for autonomous task sensing and switching during inference with multiple existing LoRA adapters embedded in a single LLM. In this work, we introduce **MeteoRA** (**M**ultiple-**t**asks **e**mbedded **LoRA**), a scalable and efficient framework that reuses multiple task-specific LoRA adapters into the base LLM via a full-mode Mixture-of-Experts (MoE) architecture. This framework also includes novel MoE forward acceleration strategies to address the efficiency challenges of traditional MoE implementations. Our evaluation, using the LlaMA2-13B and LlaMA3-8B base models equipped with 28 existing LoRA adapters through MeteoRA, demonstrates equivalent performance with the traditional PEFT method. Moreover, the LLM equipped with MeteoRA achieves superior performance in handling composite tasks, effectively solving ten sequential problems in a single inference pass, thereby demonstrating the framework's enhanced capability for timely adapter switching.

## 1 Introduction

Large language models (LLMs) have achieved significant advancement in modern intelligent applications, excelling in tasks from language comprehension to generation within the field of natural language processing (NLP) (Achiam et al., 2023; Touvron et al., 2023). By applying the fine-tuning process to pretrained LLMs, these models have demonstrated remarkable efficacy in handling domain-specific tasks. Examples include converting natural language text into SQL queries (Katsogiannis-Meimarakis & Koutrika, 2023; Pourreza & Rafiei, 2024), utilizing LLMs as agents in diverse interactive applications (Song et al., 2023; Chen et al., 2023; Gupta & Kembhavi, 2023), and developing models tailored for specific domains, such as BloombergGPT (Wu et al., 2023b) for financial analysis and ChatLaw (Cui et al., 2023) for legal consulting.

This *pretrain-fine-tune* paradigm has catalyzed the development of several parameter-efficient fine-tuning (PEFT) methods. Low-Rank Adaptation (LoRA) (Hu et al., 2021) stands out as a noteworthy exemplar of PEFT, offering efficient fine-tuning by updating only the low-rank matrices while keeping the rest of base LLM's parameters unchanged. Once fine-tuned, these matrices, which consist of a minimal number of parameters, are encapsulated as a LoRA adapter that can be readily deployed or integrated with the base LLM for enhanced functionality. To improve the capability of handling multiple tasks simultaneously, the scalability of deploying these fine-tuned LoRA adapters has been explored. Solutions such as Huggingface PEFT (Mangrulkar et al., 2022), S-LoRA (Sheng et al., 2023), and other variants have been developed to facilitate the simultaneous serving of numerous LoRA adapters on a single base LLM, enhancing the model's adaptability and efficiency in diverse application environments.

Despite the success of LoRA in the *pretrain-fine-tune* paradigm, several challenges remain. When reusing existing LoRA adapters, a primary challenge is the ability of multi-LoRA embedded LLMs to

---
*Corresponding author

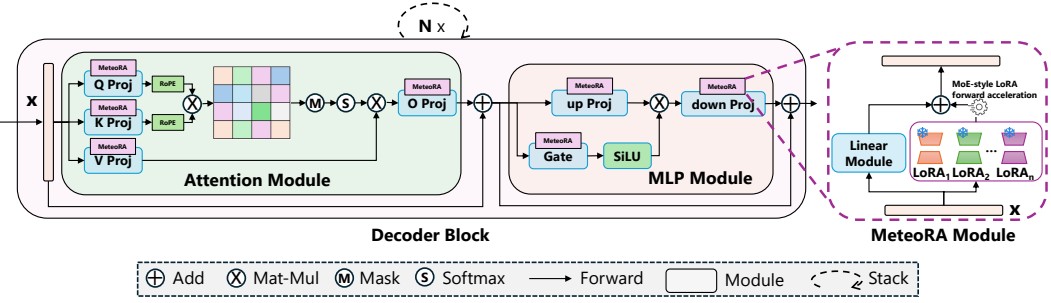

Figure 1: Our proposed framework provides a full-mode MoE architecture that directly reuses various off-the-shelf LoRA adapters, enhancing the LLM's ability to timely and autonomously activate appropriate adapters for the input. MeteoRA modules could be integrated into all basic linear layers of both Attention and MLP modules. With the MoE forward acceleration strategies, LLM equipped with MeteoRA could be capable of addressing tasks across a wide range of domains effectively.

autonomously and on-demand LoRA selection during inference, a process that should allow LLM to handle different tasks by activating the appropriate LoRA adapters without explicit user instructions. Furthermore, managing composite tasks that require timely switching between LoRA adapters presents difficulties, especially when these tasks involve multiple sub-problems each requiring specific adapter activation. Current approaches such as Huggingface PEFT and S-LoRA, while capable of serving multiple existing adapters simultaneously, mainly focus on loading rather than autonomously activating adapters, thus requiring manual intervention. Similarly, current LoRA fusion methods such as LoRAHub (Huang et al., 2023) and MoA (Feng et al., 2024), although they integrate and merge knowledge from various adapters, are not suitably designed to fuse a wide range of existing LoRA adapters with such a limited MoE framework, and lack the evidence in effectively managing dynamic adapter switching during inference for composite tasks.

In this paper, we introduce a novel multi-tasks embedded LoRA framework for LLMs to reuse existing LoRAs with the ability of autonomous task sensing and switching. The framework proposes a MoE-style module called MeteoRA. Each MeteoRA module provides a trainable Gating network with MoE forward acceleration strategies (overcome the efficiency issue in naive MoE, especially when number of experts is much larger than 8) for all LoRAs' low-rank matrices in the linear layer. As shown in Figure 1, the MeteoRA module is applicable for all kinds of layers in Transformer-based LLMs (Q, K, V, and O in attention module and up_proj, gating for SiLU (Elfwing et al., 2018), and down_proj in MLP Module). Through fine-tuning all gates with minimal resources, MeteoRA effectively integrates the existing LoRA adapters into the base LLM model with the ability of autonomously on-demand LoRA selection, without the requirements of any explicit user or system instructions. Furthermore, the presence of numerous gates[1] enhances the model with a full-mode MoE architecture, showing the capability of timely LoRA switching, addressing composite tasks with only two-shot examples as illustrations for all inputs. Our empirical evaluations, which embedded 28 existing LoRA adapters with MeteoRA to LlaMA2-13B-base and LlaMA3-8B-base, highlight the full-mode MoE capabilities and demonstrate a significant performance maintenance (e.g., MeteoRA based on LlaMA3-8B achieves only 0.4% accuracy loss when solving multiple-choice tasks.). This improvement is particularly notable in handling composite tasks, showcasing the efficacy of the MeteoRA framework. The primary contributions of MeteoRA are summarized as follows:

- **Scalable LoRA integration:** MeteoRA framework for reusing existing LoRA adapters advances the LLM's capability of autonomous on-demand LoRA selection and switching.

- **MoE forward acceleration:** revealing efficiency issue of MoE and providing the forward acceleration strategies with new GPU kernel operators to achieve a $\sim 4\times$ speedup in average while maintaining memory overhead.

---

[1]For the LlaMA3-8B model, there are 224 MeteoRA modules in total, with each of the 32 decoder layers containing 7 gates (Q, K, V, O, up_proj, gating, and down_proj).

- **Advanced performance:** Evaluation shows superior performance in composite tasks when applying MeteoRA, thereby extending the practical utility of LLMs incorporating off-the-shelf LoRA adapters.

## 2 BACKGROUND

**Low-Rank adaption.** Low-Rank Adaptation (LoRA) (Hu et al., 2021) proposes a method to reduce the number of trainable parameters required for fine-tuning in downstream tasks. LoRA injects two trainable low-rank matrices $A \in \mathbb{R}^{d \times r}$ and $B \in \mathbb{R}^{r \times h}$ into each basic linear layer's weight matrix $W \in \mathbb{R}^{d \times h}$ of the Transformer-based LLM $\mathcal{M}$. The matrix multiplication of $A$ and $B$ represents the updates $\Delta W$ to the weight matrix $W$ when fine-tuning the model. The LoRA adapter modifies the forward process of this layer as follows:

$$\boldsymbol{o} = \boldsymbol{o}_{\text{base}} + \Delta \boldsymbol{o} = \boldsymbol{x} W_{\text{base}} + \boldsymbol{x} \Delta W = \boldsymbol{x} W_{\text{base}} + ((\boldsymbol{x} \times A) \times B) \tag{1}$$

where $\boldsymbol{x} \in \mathbb{R}^d$ represents the input hidden states for any token, $A, B$ first project it to the low-rank embedding space $\mathbb{R}^r$ and then map it back to the output space $\mathbb{R}^h$. LoRA can be applied to seven types of linear layers in the Transformer: four in the self-attention module ($W_q$, $W_k$, $W_v$, and $W_o$) and three in the MLP module ($W_{\text{up\_proj}}$, $W_{\text{gating}}$, and $W_{\text{down\_proj}}$). Training LoRA adapters is straightforward. It continues to use the optimization target of causal language modeling to update LoRA's parameters while freezing the billions of parameters in the pretrained LLM $\mathcal{M}$.

**Multi-task LoRA fusion.** LoRA adapter is usually fine-tuned to a specific downstream task. To enhance the capacity of LLMs in handling multiple tasks, two paradigms are utilized in practice. One approach is to fuse datasets from different tasks and then fine-tune a single LoRA module on this combined dataset. However, Ling et al. (2024) points out the difficulty in learning all specialized knowledge of various domains in one LLM. The other approach leverages existing LoRA adapters as off-the-shelf components, directly merging these adapters into one base LLM. Current popular LoRA frameworks, such as PEFT (Mangrulkar et al., 2022) and S-LoRA (Sheng et al., 2023), allow fusing multiple LoRA adapters. However, these frameworks must explicitly assign the active injected LoRAs, leaving an obvious disadvantage of lacking autonomous on-demand LoRA selection and timely LoRA switching during inference. Existing work, such as LoRAHub (Huang et al., 2023), could combine multiple LoRA adapters without the explicit task intention given by humans. However, few-shot/in-context learning is required for LoRAHub for every single downstream task.

**Mixture-of-Experts.** MoE is a machine learning paradigm that enhances model performance and efficiency by combining predictions from multiple specialized models, or experts. Introduced by Jacobs et al. (1991), MoE uses a gating network to assign input data to the most relevant experts dynamically. This approach leverages specialized knowledge from different experts, improving overall performance on diverse and complex tasks. Recent progress, particularly by Shazeer et al. (2017), has demonstrated the effectiveness of MoE in large-scale neural networks. By using sparsely-gated MoEs, where only a subset of experts is activated for each input, computational efficiency is significantly increased without compromising model capacity. This has proven particularly useful in scaling Transformer-based architectures for various applications, such as Mixtral (Jiang et al., 2024), GLaM (Du et al., 2022), DBRX (The Mosaic Research Team, 2024) and Grok-1 (xAI, 2024).

## 3 THE PROPOSED METEORA

### 3.1 METEORA ARCHITECTURE

Given a base LLM $\mathcal{M}$ and $n$ existing LoRA adapters $\{L_1, L_2, \cdots, L_n\}$ that have already been fine-tuned with the distinct tasks $\{D_1, D_2, \cdots, D_n\}$ on $\mathcal{M}$ separately, our objective is to integrate the $n$ existing LoRA adapters into the base $\mathcal{M}$ via MeteoRA framework, resulting in a LoRA embedded model $\mathcal{M}_{\text{embed}}$. Figure 1 demonstrates the MeteoRA module complemented to each basic linear layer in LLM. The reused LoRA adapters are off-the-shelf ones, available from open-source communities or have been fine-tuned for specific tasks, Each LoRA adapter $L_i$ contains a set of low-rank matrices $\{A_i, B_i\}$. MeteoRA furnishes each basic linear layer with a wide MoE architecture to embed the low-rank matrices provided by $n$ LoRA adapters.

Figure 2 shows the architecture of MeteoRA module. To embed $n$ existing LoRA adapters, MeteoRA module leverages the MoE architecture by injecting a trainable *Gating network* $G : \mathbb{R}^d \to \mathbb{R}^n$

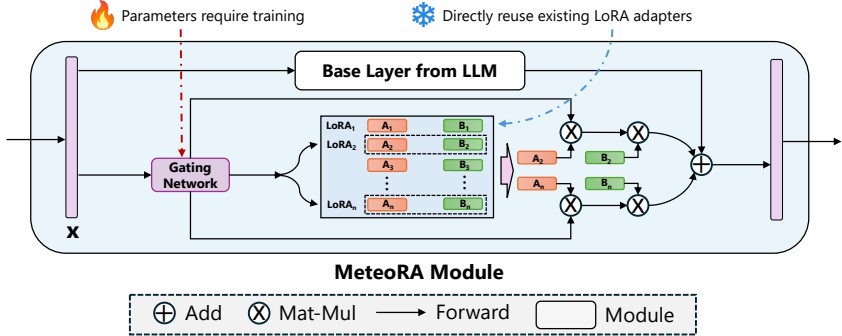

Figure 2: The architecture of MeteoRA module with MoE-style LoRA embedding. MeteoRA directly reuses existing LoRA adapters without fine-tuning and only requires training the Gating network.

together with $n$ existing pairs of $\{A_i, B_i\}$ to $\mathcal{M}$. By applying $G(\boldsymbol{x})$, MeteoRA selects $k$ pairs of $\{A_i, B_i\}$ with the top-$k$ highest gated weights for each $\boldsymbol{x}$. It then proceeds with the forward pass as follows:

$$\boldsymbol{o} = \boldsymbol{o}_{\text{base}} + \Delta\boldsymbol{o}_{I(\boldsymbol{x})} = \boldsymbol{x}W_{\text{base}} + \boldsymbol{x}\Delta W_{I(\boldsymbol{x})} = \boldsymbol{x}W_{\text{base}} + \sum_{i \in I(\boldsymbol{x})} w_i \cdot ((\boldsymbol{x} \times A_i) \times B_i) \quad (2)$$

where $I(\boldsymbol{x}) := \{i_1, i_2, .., i_k\}$ denotes the top-$k$ LoRAs selected for each token $\boldsymbol{x}$, which may varies from one another in every batch, and $w_i$ is the normalized weight of the selected LoRA $L_i$. $w_i$ could be calculated as follows:

$$w_i = \text{softmax}(G_i(\boldsymbol{x})) = \frac{\exp(G_i(\boldsymbol{x}))}{\sum\limits_{j \in I(\boldsymbol{x})} \exp(G_j(\boldsymbol{x}))} \quad (3)$$

where $G_i(\boldsymbol{x})$ denotes the unnormalized gated *logits* for the $i$-th LoRAs. By doing this, the Gating network performs as a routing strategy for selecting the appropriate LoRA adapters based on the layer's input. Each MeteoRA module contains a Gating network, and the Gating networks from different MeteoRA modules make decisions based on their own inputs, the selection of LoRA adapters could be dynamically switched in the forward process of each MeteoRA module through all LLM's decoder blocks. MeteoRA also applies top-1 and top-$k$ gating strategies as detailed in Appendix A.1.

## 3.2 LEARNING ALGORITHM

Training the injected MeteoRA modules adheres to the principles of fine-tuning LLM under autoregressive language modeling tasks. Given that $n$ pre-trained LoRA adapters, the training procedure for MeteoRA needs to maintain the parameters of the base LLM $\mathcal{M}$ and the $n$ LoRA adapters fixed. Since MeteoRA supports top-$k$ experts (LoRAs) selection, we introduce the joint optimization that combines the loss of autoregressive language modeling $\mathcal{L}_{\text{lm}}$ and all losses of Gating networks $\mathcal{L}_{\text{gate}}$:

$$\mathcal{L} = \mathcal{L}_{\text{lm}} + \beta\mathcal{L}_{\text{gate}} = \arg\max_\theta \sum_{i=1}^{L} (\log P(x_i \mid x_{i-1}; \theta) + \beta \sum_{j=1}^{B} \sum_{k=1}^{m} l_{k,j}(h)) \quad (4)$$

where $\beta$ is the hyper-parameter, $i$ is the token index, $L$ is the length of the language sequence represented as tokens, $x_i$ represents the token. The loss $l_{k,j}$ is the cross-entropy loss for LoRA classification in one MeteoRA module. For a base $\mathcal{M}$ contained $B$ decoder blocks with $m$ MeteoRA modules in each decoder, $\mathcal{L}_{\text{gate}}$ sums the loss $l_{k,j}(h)$ based on the corresponding hidden inputs $h$.

## 3.3 METEORA FORWARD ACCELERATION

The core component of the MeteoRA module is a MoE architecture that incorporates $n$ existing LoRA adapters. The classic MoE forward method, called *loop-original*, employs a for-loop style of computation that processes only the tokens assigned to the $i$-th LoRA adapter in the $i$-th iteration,

leading to inefficiency especially when token-adapter assignments are sparse or when $b \times s < n$ (e.g., during decoding inference phase where $s$ is fixed to 1), and resulting in up to a $\mathbf{10\times}$ **slowdown** compared to the *single-lora* forward in our experiments. To address this, we introduce *bmm-torch* method which parallelizes the computation by performing batched matrix multiplications (BMM) (PyTorch, 2024) for all $b \times s \times k$ tokens and $n$ adapters at once, represented by:

$$\underbrace{[\Delta o_1, \ldots, \Delta o_{bs}]}_{b \times s} = \sum_k \underbrace{[w_1, \ldots, w_{bsk}]}_{b \times s \times k} \odot \left( \left( \underbrace{[x_1, \ldots, x_{bsk}]}_{b \times s \times k} \times \underbrace{[A_{i_1}, \ldots, A_{i_{bsk}}]}_{b \times s \times k} \right) \times \underbrace{[B_{i_1}, \ldots, B_{i_{bsk}}]}_{b \times s \times k} \right)$$

This results in a $\mathbf{4\times}$ **speedup** over *loop-original*, thus only $\sim \mathbf{2.5\times}$ slower than the *single-lora* in most of our experiments (see Section 4.4). However, *bmm-torch* achieves its great efficiency by temporarily allocating a $\frac{b \times s \times k}{n}$-sized space on the HBM for batched $\mathcal{A}$, $\mathcal{B}$, causing a potential out-of-memory risk for those tasks when either $b$ or $s$ is quite large. To mitigate this, we further propose *bmm-triton* method with a custom GPU kernel implemented in Triton (Tillet et al., 2019) to resolve this memory issue. In our evaluation, *bmm-triton* achieves $\sim 80\%$ performance of *bmm-torch* while maintaining the same low memory footprint as *loop-original* (see Section 4.4). This makes *bmm-triton* a more suitable solution for large-scale tasks, effectively balancing computational speed and memory efficiency. Thus, the two proposed acceleration methods could together boost the inference in practice, by using *bmm-triton* in the prefill phase (where the sequence length $s$ varies based on the input sequence) and *bmm-torch* in the decoding phase (where the sequence length $s = 1$). The details on the design of the *bmm-triton* kernel are provided in Appendix A.2.

## 4 EVALUATION

We conduct experiments on individual and composite tasks as detailed in Section 4.1. For our base models, we use two well-known LLMs, LlaMA2-13B (Touvron et al., 2023) and LlaMA3-8B (Meta, 2024). The code and the models are available[2].

### 4.1 EVALUATION SETTINGS

**LoRA tasks and datasets.** We select 28 tasks from well-known benchmarks for our experiment. Specifically, our task set consists of 22 tasks from BigBench (bench authors, 2023), three non-English to English translation tasks from News-Commentary (Tiedemann, 2012), and three widely utilized tasks: GSM8K (Cobbe et al., 2021), CNN/DailyMail (See et al., 2017), and Alpaca (Taori et al., 2023). These 28 tasks span a variety of NLP categories, such as contextual comprehension, conversational question answering, summarization, translation, mathematics, logical reasoning, and multilingual challenges. For detailed task descriptions, refer to Appendix A.3.

**Metrics.** We apply a zero-shot evaluation setting for all tasks, adding brief task descriptions for tasks such as CNN/DailyMail and the three translation tasks that do not inherently include task descriptions. As for metrics, we use *accuracy* for multiple-choice tasks and GSM8K while employing metrics such as *BLEU*, *ROUGE-1*, *ROUGE-2*, and *ROUGE-L* for other tasks.

**Models.** We use LlaMA2-13B and LlaMA3-8B as the base LLMs for LoRA and MeteoRA adaption. Both LlaMA models are pretrained LLMs and do not include the process of instruction tuning. We train specific LoRA adapters for each task using their respective training sets. The process of training LoRA adapters could be offline or dismissed when off-the-shelf LoRA is accessible. Then, the Gating networks, which embed the adapters in the MeteoRA module, are fine-tuned efficiently based on the balanced dataset containing 1,000 samples for each task. The Gating networks for 28 tasks take no more than 10 hours to reach the convergence with 4 H800 training via Accelerate (Gugger et al., 2022). For scenarios where the training data for the original LoRA adapter is limited, we train Gating networks using a top-2 strategy, with only 100 and 5 samples accessible per task.

---

[2]The implementation code is accessible at `https://github.com/NJUDeepEngine/meteora`, and the two MeteoRA embedded LLMs are available at`https://huggingface.co/NJUDeepEngine/MeteoRA-llama2-13b` and `https://huggingface.co/NJUDeepEngine/MeteoRA-llama3-8b`

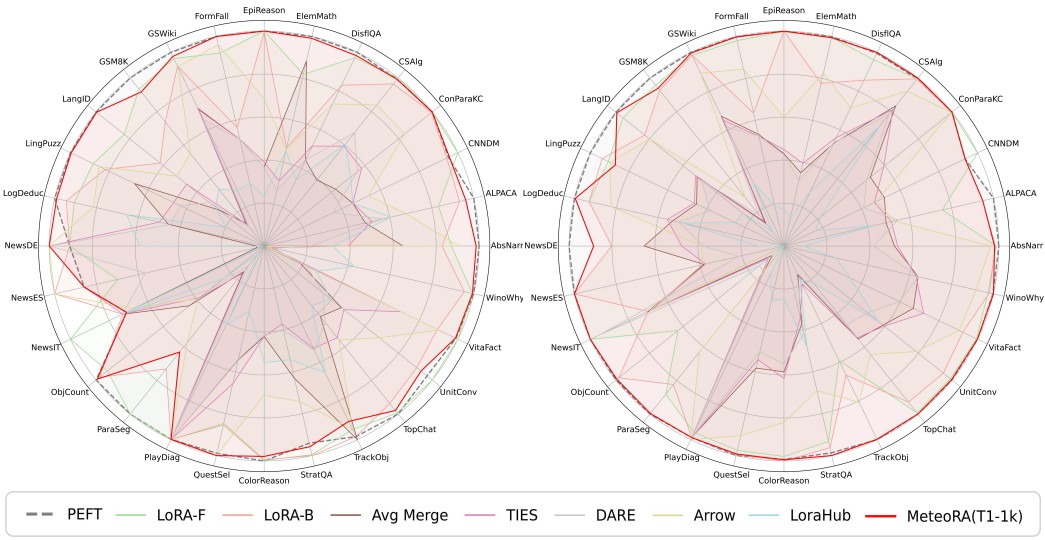

Figure 3: Evaluation results on the 28 selected tasks. The results on the left are based on LlaMA2-3B, while those on the right are based on LlaMA3-8B. The MeteoRA performs similarly on most tasks, leading to high overlap between the two polygons in the radar graphs. For clarity, we only draw results from MeteoRA with top-1 strategy in the radar graphs. Detailed results for each individual task are available in Appendix A.4

For baseline comparisons, we train one LoRA adapter (i.e., LoRA-F) using a mixed training set from all 28 tasks, and another LoRA adapter (i.e., LoRA-B) with the balanced dataset designed for training the Gating network. We also use Huggingface PEFT (short in PEFT) loading all 28 LoRA adapters (same ones used for MeteoRA) with explicit LoRA activation information during evaluation as a reference model. Additionally, we include several LoRA merge methods for comparison, including: averaging 28 LoRA adapters (referred to as Avg Merge), TIES (Yadav et al., 2024a), DARE (Yu et al., 2024), Arrow (Ostapenko et al., 2024), and LoraHub (Huang et al., 2023).

All LoRA adapters interact with all seven linear layers in LLaMA's Decoder layer, configured with $r = 8$, $\alpha = 16$, and a learning rate of $5e - 5$. Due to some tasks having small training sets, the batch size for fine-tuning is set to $4$. All our experiments were conducted on a GPU server with five H800 80G GPUs. Notice that we carefully selected the training hyperparameters for the LoRA-F and LoRA-B to ensure that their performance on the 28 tasks would not be excessively incomparable.

**Composite tasks.** To evaluate the model's ability to sequentially solve composite tasks, we construct three composite evaluation sets by serially concatenating independent tasks. These evaluation sets, referred to as *composite-3*, *composite-5*, and *composite-10*, consist of 3, 5, and 10 tasks, respectively, each containing 200 samples. The samples in each *composite-n* set can be viewed as a single "paper sheet" created by concatenating the n tasks in sequence. During evaluation, the entire "paper sheet" is input into the model, which is required to sequentially generate both the task number and the corresponding answer for each task in the order presented. This setup tests the model's ability to handle multiple tasks within a single input, maintaining coherence across the sequence. Temperature scaling is involved in Gating network. More details refer to Figure 4, Appendix A.5 and A.6.

## 4.2 MAIN RESULTS

Figures 3 demonstrates the performance of the MeteoRA models, LoRA-F, LoRA-B, 5 LoRA merge methods, and a reference model PEFT based on LlaMA2-13B and LlaMA3-8B, respectively, across the selected 28 tasks. Table 1 shows the averaged scores in various matrics for all methods.

The evaluation results indicate that, regardless of the base LLM, the MeteoRA models utilizing the top-1 strategy achieve performance very close to the reference model PEFT, while no explicit LoRA activation/deactivation is required in MeteoRA. Although LLMs with both LoRA-F and LoRA-B reach comparable performance on several certain tasks, they exhibit significantly poorer outcomes on

Table 1: Results of the 28 selected tasks on LlaMA2-13B/LlaMA3-8B base LLMs. T1 and T2 represent the top-1 and top-2 strategies, while the subsequent numbers indicate the number of accessible samples per task for gate training. Our methods perform the best in most tasks. Notice that the task *linguistics_puzzles* achieves significantly higher ROUGE scores on LlaMA3-8B base, disproportionately influencing the average ROUGE scores and resulting in slightly higher averages for LoRA-B. Excluding this outlier, our methods consistently lead in performance across the evaluation.

| Model | Accuracy↑ | BLEU↑ | ROUGE-1↑ | ROUGE-2↑ | ROUGE-L↑ |
|---|---|---|---|---|---|
| PEFT (reference) | 0.762 / 0.817 | 35.66 / 45.32 | 0.340 / 0.341 | 0.163 / 0.164 | 0.316 / 0.317 |
| LoRA-F | 0.730 / 0.767 | 41.27 / 42.93 | 0.318 / 0.327 | 0.136 / 0.157 | 0.294 / 0.306 |
| LoRA-B | 0.666 / 0.750 | 37.98 / 38.47 | 0.314 / **0.343** | 0.128 / **0.171** | 0.288 / **0.321** |
| Avg Merge | 0.370 / 0.427 | 19.23 / 39.89 | 0.231 / 0.200 | 0.082 / 0.060 | 0.184 / 0.158 |
| TIES | 0.388 / 0.441 | **47.28** / 34.66 | 0.195 / 0.199 | 0.055 / 0.059 | 0.151 / 0.158 |
| DARE | 0.332 / 0.404 | 46.53 / 36.74 | 0.192 / 0.188 | 0.054 / 0.056 | 0.144 / 0.147 |
| Arrow | 0.569 / 0.647 | 41.03 / 29.93 | 0.281 / 0.283 | 0.123 / 0.142 | 0.234 / 0.242 |
| LoraHub | 0.307 / 0.235 | 13.43 / 10.11 | 0.158 / 0.141 | 0.049 / 0.035 | 0.124 / 0.104 |
| MeteoRA (T1-1k) | 0.755 / **0.811** | 36.73 / **45.64** | **0.336** / 0.338 | 0.160 / 0.158 | 0.313 / 0.314 |
| MeteoRA (T2-1k) | 0.748 / 0.806 | 38.97 / 44.98 | **0.336** / 0.337 | **0.161** / 0.158 | **0.314** / 0.313 |
| MeteoRA (T2-100) | **0.758** / 0.783 | 39.44 / 39.90 | 0.331 / 0.309 | 0.159 / 0.139 | 0.281 / 0.256 |
| MeteoRA (T2-5) | 0.740 / 0.773 | 38.37 / 40.12 | 0.328 / 0.299 | 0.156 / 0.131 | 0.277 / 0.246 |

Table 2: The evaluation results of *composite-n* tasks. MeteoRA is marked in color on the left side, while LoRA-B is in black on the right side. Refer to Appendix A.5 for a detailed explanation.

| Metric | composite-3 | | composite-5 | | composite-10 | |
|---|---|---|---|---|---|---|
| # Avg Attempt | 2.95↓ | 3.00 | 4.63↑ | 4.33 | 8.24↑ | 6.07 |
| # Avg Correct | 1.49↑ | 1.31 | 2.62↑ | 2.42 | 3.75↑ | 2.95 |
| Avg BLEU | 15.31↑ | 10.55 | 9.86↑ | 9.41 | 8.85↑ | 8.71 |
| Avg ROUGE-1 | 0.195↑ | 0.135 | 0.221↑ | 0.219 | 0.238↑ | 0.161 |
| Avg ROUGE-2 | 0.052↑ | 0.027 | 0.069↑ | 0.063 | 0.059↑ | 0.043 |
| Avg ROUGE-L | 0.182↑ | 0.128 | 0.207↓ | 0.208 | 0.209↑ | 0.123 |

others. Additionally, MeteoRA employing the top-2 strategy, despite occasionally showing greater capability loss compared to MeteoRA with top-1 strategy, occasionally outperforms PEFT with adapters trained directly on the individual tasks. This suggests that the $L_{lm}$ component in the loss function (Equation 4) becomes influential in these cases, indicating a beneficial mix of LoRA adapters from various tasks for future study. For the MeteoRA (T2-100) and MeteoRA (T2-5), although their performance shows a gap compared to MeteoRA 1k, they still outperform the baseline models on most metrics. This demonstrates that the Gating network can still learn to effectively utilize existing LoRA adapters with only a few examples.

## 4.3 COMPOSTE-N TASKS

The evaluation results for these three tasks are illustrated in Table 2. Notice that only LlaMA3-8B with the MeteoRA (top-2 strategy) and LoRA-B effectively address these *composite-n* tasks. Subsequent discussions will therefore focus exclusively on these two models. Although the MeteoRA model attempts slightly fewer questions than LoRA-B in *composite-3* tasks, it correctly answers a higher number of multiple-choice questions and achieves superior BLEU and ROUGE scores. As the task complexity increases to *composite-5* and *composite-10*, MeteoRA outperforms LoRA-B in almost all metrics. For more details, refer to Appendix A.5.

To further validate the functionality of the Gating network in the MeteoRA block, we display the LoRA selection patterns in the inference process of a *composite-3* sample in Figure 4. With the top-2 strategy, Gating network appropriately assigns greater weight to the corresponding LoRA adapters for the majority of the tokens, no matter in input or output. At the junctions of two adjacent tasks, the Gating network correctly performed the timely switching actions of LoRA adapters.

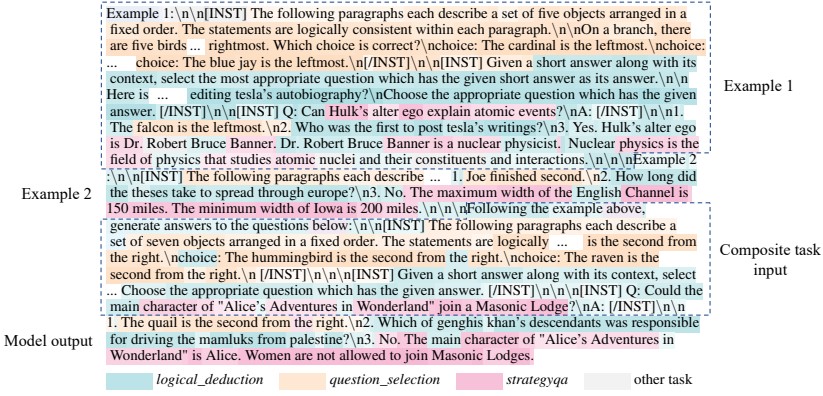

Figure 4: An example of *composite-3* task. We highlight the statistically dominant LoRA selected by MeteoRA in token level (decoded to words). The result shows that LLM with MeteoRA could achieve timely LoRA switching on both phases of input understanding and output generation. The background color gets darker when Gating network assigns a higher weight value.

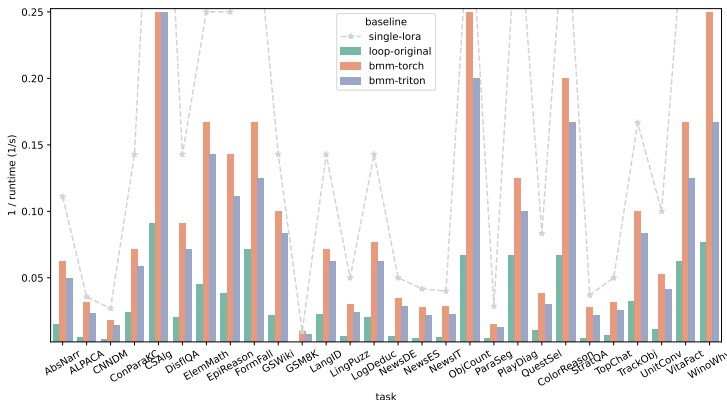

Figure 5: The overall *root-of-runtime* of *four* forward pass designs on 28 different Big-Bench subtasks.

## 4.4 EFFICIENCY

To assess the efficiency of our novel forward pass designs using custom GPU kernel operators, we truncate $batch\_size \times 10$ samples from each test dataset of all 28 tasks. We evaluate these designs alongside *four* variants with the same hyperparameters: the upper-bound *single-lora*, the baseline *loop-original*, and two novel forward acceleration strategies based on $bmm$: *bmm-torch* and *bmm-triton*, implemented by PyTorch and Triton respectively. Figure 5 displays the histogram of the overall *root-of-runtime* metric for each task and design. Additional evaluation is detailed in Appendix A.7.

## 5 RELATED WORK

**Multi-task fusion.** Our proposed method falls into the field of LoRA adapter composition for multi-task fusion. The first category focuses on fusing the entire models. Researchers mainly study model ensembling and multi-task learning to achieve this goal. Existing works integrate the models under the setting of shared model architecture (Matena & Raffel, 2022; Jin et al., 2022; Wu et al., 2023a; Yadav et al., 2024b). Others focus on merging models with various architectures or from different tasks. Both methods (Stoica et al., 2023; Liu et al., 2022) try to merge models that are trained for various tasks without additional training. The second category is more concerned with fusion in terms of the tasks. Ilharco et al. (2022) proposes a model editing method via task vectors.

Sun et al. (2022) leverage in-context learning with few-shots to enhance the performance of unseen tasks. However, these methods require multi-task training or prior knowledge for the evaluation tasks. Our method embeds off-the-shelf LoRA adapters with a Gate network in the MeteoRA module. None of the examples (zero-shot) are required for all individual tasks.

**Fusion under MoE**. In the context of *pretrain-fine-tune* paradigm, PEFT becomes a common sense for developing Transformer-based LLM downstream applications. Directly fine-tuning on a fused dataset from various tasks is unable to achieve better performance Ling et al. (2024). Some works focus on leveraging existing LoRA adapters as off-the-shelf components, integrating them directly into a base LLM. For example, PEFT (Mangrulkar et al., 2022) and S-LoRA (Sheng et al., 2023) are frameworks aiming to embed multiple LoRA adapters to one LLM. However, requiring explicit activation/deactivition during usage. MixLoRA (Li et al., 2024) targets to a resource-efficient sparse MoE model, fine-tuning MoE on MLP module with the auxiliary load balance loss used in Mixtral (Jiang et al., 2024). Although MixLoRA supports LoRA adapters for the attention layer, the adapters are still dense models encompassed with the linear layers in the attention module. Others (Huang et al., 2023; Yang et al., 2024; Feng et al., 2024; Chen et al., 2024; Wu et al., 2023c) propose LoRA fusion based on the concept of Mixture-of-experts that enhance the model's ability for cross-domain tasks. However, the methods mainly focus on fusing LoRA adapters to the FFN module or Q in the attention module. Our method could embed all kinds of LoRA adapters. By leveraging the full-mode MoE architecture, the LLM's capacity could be boosted with autonomous and timely LoRA switching, especially for solving composite tasks.

# 6 LIMITATIONS

**LoRA adapter update.** Although the Gating network within MeteoRA module is trained separately among the adapters, it is necessary to retrain or fine-tune the Gating network if some LoRA adapters are updated. The Gating network is trained using the hidden state as inputs, which are influenced by LoRA adapters in previous layers. Testing revealed that directly replacing some LoRA adapters with improved versions did not enhance performance on our test set. However, after retraining the MeteoRA modules, the LLM equipped with MeteoRA exhibited performance improvements. Technically, this issue may be related to the domain shift problem, where the Gating network is applied to another operational field the distribution shift. Employing statistical methods such as (Li et al., 2020; Krishnan & Tickoo, 2020) may help calibrate the output of the Gating network to produce more accurate logits and results.

**Knowledge fusion tasks.** Composite tasks, which involve a broad range of tasks, represent one type of complexity in terms of the scope of tasks. More challenging are tasks that require knowledge fusion across domains. To assess the capability of MeteoRA in knowledge fusion task, we construct a mathematics task by translating problems from GSM8K into a foreign language (e.g., Italian), so that the LLM with MeteoRA must solve these foreign language GSM8K problems by leveraging knowledge from both GSM8k LoRA (trained on problems in English) and the foreign language LoRA (trained for Italian to English translation). Although MeteoRA successfully fuses the two LoRA adapters to address the math problems in a foreign language, it does not show superior performance compared to LLM equipped only with the GSM8K LoRA. We hypothesize that the base LLM's existing proficiency in the selected foreign language may render the additional adapter unnecessary. Future efforts could focus on constructing more suitably complex tasks where the required cross-domain knowledge is not already pre-trained into the base LLM.

**MoE efficiency.** Sparsely-gated MoE (Shazeer et al., 2017) offers computational efficiency advantages over dense MoE. However, the naive implementation of MoE forward (*loop-original*), such as the SparseMoE in Mixtral (Jiang et al., 2024; Wolf et al., 2020), still encounters efficiency issues when the number of experts increases. In our evaluations, the runtime for inference can be up to t longer than that of *single-lora* when embedding 28 LoRA adapters into one LLM. With our proposed forward acceleration techniques *bmm-torch* and *bmm-triton*, we achieve a speedup of $\sim 4\times$ compared to the *loop-original*, though this still falls short of the ideal upper bound (*single-lora*). Technically, it is extremely difficult to increase the inference speeds for MeteoRA when the number of embedded LoRA adapters increases. Future work could explore developing new operators in triton or CUDA to continuously enhance MoE acceleration in terms of memory efficiency.

# 7 CONCLUSIONS

This paper presents a framework MeteoRA that achieves scalable multi-task LoRA embedding within LLMs, enhancing the existing LLMs with a full-mode MoE architecture with forward acceleration strategies. LLMs equipped with MeteoRA enhance the ability to autonomously select the most pertinent LoRA adapters to generate appropriate responses. Moreover, its capability for timely LoRA switching leads to superior performance, particularly in sequentially solving composite tasks. Future work could explore the transformative potential of MeteoRA in multifaceted problem-solving scenarios, and inference efficiency by designing more efficient GPU kernel operators.

## ACKNOWLEDGMENT

We are thankful to the anonymous reviewers for their helpful comments. This work is supported by Frontier Technologies R&D Program of Jiangsu (#BF2024059), the National Natural Science Foundation of China (Grants #62172199), and the Collaborative Innovation Center of Novel Software Technology and Industrialization. Jingwei Xu (`jingwei@nju.edu.cn`) is the corresponding author.

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

## A APPENDIX

### A.1 TOP-K STRATEGY

**Top-**$1$ **strategy:** When the Gating network is configured to select the LoRA adapter with the maximum logit, the forward process of MeteoRA as detailed in Equation 2 simplifies to the classical LoRA forward $o = \boldsymbol{x}W_{\text{base}} + (\boldsymbol{x} \times A_i) \times B_i$. Thus, the weight $w_i$ calculated by the Gating network $G_i$ only contributes to the LoRA selection, but does not influence the token generation process, resulting in it being irrelevant to the loss $\mathcal{L}_{\text{lm}}$. Thus, training of Gating networks under the top-1 strategy could utilize the following truncated loss function $\mathcal{L}_{\text{top}-1}$:

$$\mathcal{L}_{\text{top}-1} = \arg\max_\theta \sum_{i=1}^{L} \sum_{j=1}^{B} \sum_{k=1}^{m} l_{k,j}(h) \tag{5}$$

**Top-**$k$ **strategy:** With the top-$k$ strategy set in the Gating network, MeteoRA computes the normalized weights $w_i$ for the $k$ selected LoRA adapters. These weights participate in the computation of the losses of both $\mathcal{L}_{\text{lm}}$ and $\mathcal{L}_{\text{gate}}$ as specified in Equation 2. Thus, the parameter updates for the Gating network derive from the losses associated with both LoRA classification and autoregressive token generation. Notice that the LoRA classification loss is only influenced by the LoRA adapter with the highest logit, whereas the backpropagation from the token loss affects the parameters in the Gating network responsible for all $k$ selected LoRA adapters. Although these remaining $k - 1$ adapters lack direct supervision from the LoRA classification loss, the token generation loss contributes to enhanced robustness and the capacity for LoRA switching during generation.

### A.2 DETAILS ON THE TAILORED TRITON KERNEL FOR EFFICIENT METEORA FORWARD

To address the memory copying problem caused by PyTorch indexing, we fuse the two $bmm$ operations inside a GPU kernel function implemented by Triton, which dynamically indexes the right pair of LoRA matrix $(A_i, B_i)$ and load them from HBM to SRAM in each parallelized thread. Therefore, there is no need to explicitly allocate $b \times s \times k$ pairs of $(A_i, B_i)$ over the original $n$ ones.

Another challenge is that Triton constraints all the dimensions for the matrix operators should be no less than 16, however, under the MeteoRA settings, this requirement can never be satisfied since the first operator $\boldsymbol{x}$ is a vector, and also, the LoRA rank size may be less than 16 easily (e.g., in all our experiments, we fix $r = 8$). Therefore, it is not that trivial to implement such a kernel, unless using the simple *masking* strategy to meet the requirements with over $15\times$ waste of I/O.

---

**Algorithm 1** Pseudo Code for BMM-Triton Kernel Function

---

1: Prepare blockized $X, A'$ with their masks $M_1, M_2$ before launching the kernel
2: Load $X, I, M_1, M_2$ from HBM to SRAM             $\triangleright$ $I$ is the candidate LoRA index set
3: Load $A', B$ indexed by $I$
4: $oA' = X \times A'$
5: $oA'' \leftarrow \big((oA' \odot M_1) \times M_2\big)$
6: $oB' \leftarrow oA'' \times B$
7: $O \leftarrow \text{colsum}[oB']$                           $\triangleright$ Compute column-wise sum
8: Store $O$ back from SRAM to HBM

---

To both obey the dimension constraint and avoid too much waste, for the first $bmm$ of $\boldsymbol{x} \in \mathbb{R}^{1 \times d}$ and $A \in \mathbb{R}^{d \times r}$, we use a *blocking* strategy to split the vector $\boldsymbol{x}$ along the hidden size dimension by $m$ blocks, where $m >= 16$. In such case, the first operator becomes a matrix $X$ with shape $(m, \frac{d}{m})$, and also we have to split $A$ along the first dimension to become a more square matrix $A'$ with shape $(\frac{d}{m}, r \times m)$. Notice that now the output of first $bmm$: $oA' = X \times A'$ with shape $(m, r \times m)$ has a relationship with the original one $oA = \boldsymbol{x} \times A$ with shape $(1, r)$ as follows:

$$oA'' = \big((oA' \odot M_1) \times M_2\big)$$
$$oA = \text{colsum}[oA''] \tag{6}$$

where $M_1$ and $M_2$ are two trivial 01 mask matrixs, each sized $(m, r \times m)$ and $(r \times m, r)$ respectively. So we can transform back to the right results by three additional negligible dot-product with $M_1$, matrix-product with $M_2$, and colsum operations for the first $bmm$. For the second one, instead of directly using the right result $oA$, we can delay the colsum operation until we finish the second $bmm$, i.e. we use $oA''$ with shape $(m, r)$ and $B$ with shape $(r, h)$ to do matrix-product operations to get the temporary result $oB'$ with shape $(m, h)$, then apply colsum to get the final LoRA output $O$ with shape $(1, h)$. Notably, on one hand, we can avoid one more *blocking* operation for $oA$ since $oA''$ already meets the dimension constraint, on the other hand, if $r < 16$, we can just simply utilize *masking* strategy since it is the inner dimension and small enough.

Overall, for the Triton kernel function, we offer the pseudo code as shown in Algorithm 1.

## A.3 INFORMATION ABOUT 28 TASKS

Table 3 shows the detailed information of the 28 selected tasks in the Section 4. The name in parentheses is the abbreviation of the corresponding task. We use the original training sets from these tasks to fine-tune the LoRA adapters and Gating networks in MeteoRA modules. To achieve a balanced fine-tuning across the diverse task spectrum and ensure efficient training, we construct a balanced dataset by randomly sampling 1,000 samples from each task. This balanced dataset is then divided into a training set with 25,200 samples (i.e., 900 samples for each task) and a validating set with 2,800 samples (i.e., 100 samples for each task) for fine-tuning. In terms of the evaluation part, the performances are evaluated on each task's original test set.

Table 3: Details about the 28 selected tasks.

| Task Name | Keywords | Description | Evaluation Metrics |
|---|---|---|---|
| abstract_narrative_understanding (AbsNarr) | narrative understanding, multiple choice | Given a narrative, choose the most related proverb. | Accuracy |
| alpaca (ALPACA) | instruction-tuning | Write appropriate answers according to instructions. | BLEU, ROUGE |
| cnn_dailymail (CNNDM) | summarization | Given news articles, write the summarization. | ROUGE |
| contextual_parametric_knowledge_conflicts (ConParaKC) | contextual question-answering, multiple choice | Answer questions given the contextual information. | Accuracy |
| cs_algorithms (CSAlg) | algorithms, numerical response | Solve two common computer-science tasks. | Accuracy |
| disfl_qa (DisflQA) | contextual question-answering, reading comprehension | Pick the correct answer span from the context given the disfluent question. | Accuracy |
| elementary_math_qa (ElemMath) | mathematics | Answer multiple choice mathematical word problems. | Accuracy |
| epistemic_reasoning (EpiReason) | logical reasoning, multiple choice | Determine whether one sentence entails the next. | Accuracy |
| formal_fallacies_syllogisms_negation (FormFall) | logical reasoning, multiple choice, | Distinguish deductively valid arguments from formal fallacies. | Accuracy |
| goal_step_wikihow (GSWiki) | causal reasoning, multiple choice | Perform one of three subtasks: step inference, goal inference, or step ordering. | Accuracy |
| gsm8k (GSM8K) | mathematics | Solve the grade school math word problems. | Accuracy |
| language_identification (LangID) | multilingual, multiple choice | Given a sentence, select the correct language. | Accuracy |
| linguistics_puzzles (LingPuzz) | logical reasoning, linguistics | Solve Rosetta Stone-style linguistics puzzles. | BLEU, ROUGE |
| logical_deduction (LogDeduc) | logical reasoning, multiple choice | Deduce the order of a sequence of objects. | Accuracy |
| news_commentary_de (NewsDE) | multilingual, translation | Translate German sentences into English. | BLEU |
| news_commentary_es (NewsES) | multilingual, translation | Translate Spanish sentences into English. | BLEU |
| news_commentary_it (NewsIT) | multilingual, translation | Translate Italian sentences into English. | BLEU |
| object_counting (ObjCount) | logical reasoning | Questions that involve enumerating objects and asking the model to count them. | Accuracy |
| paragraph_segmentation (ParaSeg) | segmentation, multilingual | Identify the sentences that end a paragraph in a document. | Accuracy |
| play_dialog_same_or_different (PlayDiag) | reading comprehension, multiple choice | Determine if nearby lines in a Shakespeare play were spoken by the same individual. | Accuracy |
| question_selection (QuestSel) | reading comprehension, multiple choice | Given an answer along with its context, select the most appropriate question which has the given answer as its answer. | Accuracy |
| reasoning_about_colored_objects (ColorReason) | reading comprehension, logical reasoning, multiple choice | Answer extremely simple questions about the colors of objects on a surface. | Accuracy |
| strategyqa (StratQA) | logical reasoning, context-free question answering | Answer questions in which the required reasoning steps are implicit in the question. | BLEU, ROUGE, Accuracy |
| topical_chat (TopChat) | free response | Open-domain response generation. | BLEU, ROUGE |
| tracking_shuffled_objects (TrackObj) | logical reasoning, multiple choice | Determine the final positions given initial positions and a description of a sequence of swaps. | Accuracy |
| unit_conversion (UnitConv) | contextual question-answering, mathematics, multiple choice | Perform various tasks relating to units, including identification and conversion. | Accuracy |
| vitaminc_fact_verification (VitaFact) | truthfulness, reading comprehension, multiple choice | Identify whether a claim is True or False based on the given context. | Accuracy |
| winowhy (WinoWhy) | causal reasoning, multiple choice | Evaluate the reasoning in answering Winograd Schema Challenge questions. | Accuracy |

## A.4 EXPERIMENTAL RESULTS OF 28 TASKS

Table 4, Table 5, Table 6 and Table 7 show the detailed evaluation results of different models on the 28 selected tasks. When drawing Figure 3, for tasks we use BLEU and ROUGE as metrics, we selected BLEU for *news_commentary_de*, *news_commentary_es*, and *news_commentary_it*, while opting for ROUGE-L for the remaining tasks.

Table 4: Experimental results for tasks using accuracy as metric (LlaMA2-13B base model).

| Task Name | PEFT (reference) | LoRA-F | LoRA-B | Avg LoRA | TIES | DARE | Arrow | LoraHub | MeteoRA (T1-1k) | MeteoRA (T2-1k) | MeteoRA(T2-100) | MeteoRA(T2-5) |
|---|---|---|---|---|---|---|---|---|---|---|---|---|
| AbsNarr | 0.863 | 0.758 | 0.720 | 0.562 | 0.340 | 0.190 | 0.788 | 0.278 | 0.858 | 0.860 | 0.860 | **0.868** |
| ConParaKC | 0.999 | 0.999 | 0.994 | 0.424 | 0.579 | 0.554 | 0.836 | 0.514 | **0.999** | **0.999** | **0.999** | 0.998 |
| CSAlg | 0.841 | **0.848** | 0.818 | 0.333 | 0.504 | 0.572 | 0.712 | 0.515 | 0.841 | 0.818 | 0.826 | 0.826 |
| DisflQA | 0.690 | 0.670 | 0.573 | 0.306 | 0.356 | 0.307 | 0.506 | 0.236 | 0.679 | **0.684** | 0.683 | 0.661 |
| ElemMath | 0.801 | 0.671 | 0.375 | 0.707 | 0.249 | 0.212 | 0.369 | 0.364 | **0.794** | 0.725 | 0.771 | 0.718 |
| EpiReason | 1.000 | **1.000** | 0.995 | 0.367 | 0.390 | 0.367 | 0.685 | 0.233 | **1.000** | 0.998 | **1.000** | **1.000** |
| FormFall | 0.999 | 0.921 | 0.565 | 0.510 | 0.510 | 0.510 | 0.961 | 0.299 | 0.999 | 0.996 | **1.000** | 0.999 |
| GSWiki | 0.906 | 0.877 | 0.842 | 0.639 | 0.646 | 0.591 | 0.839 | 0.260 | **0.887** | 0.872 | 0.879 | 0.881 |
| GSM8K | 0.458 | 0.428 | 0.338 | 0.062 | 0.058 | 0.052 | 0.252 | 0.155 | 0.420 | **0.439** | 0.427 | 0.397 |
| LangID | 0.874 | 0.728 | 0.542 | 0.235 | 0.403 | 0.283 | 0.455 | 0.253 | **0.872** | 0.854 | 0.869 | 0.848 |
| LogDeduc | 0.720 | 0.653 | 0.680 | 0.330 | 0.360 | 0.323 | 0.587 | 0.473 | 0.713 | 0.717 | 0.720 | **0.723** |
| ObjCount | 0.740 | 0.690 | 0.725 | 0.330 | 0.285 | 0.245 | 0.290 | 0.180 | 0.735 | 0.725 | **0.740** | 0.720 |
| ParaSeg | 0.214 | 0.274 | 0.214 | 0.047 | 0.050 | 0.036 | 0.178 | 0.015 | 0.195 | 0.182 | **0.297** | 0.295 |
| PlayDiag | 0.649 | 0.649 | **0.650** | 0.649 | 0.649 | 0.649 | 0.649 | 0.265 | 0.649 | 0.649 | 0.649 | 0.649 |
| QuestSel | 0.927 | 0.801 | 0.794 | 0.509 | 0.617 | 0.506 | **0.937** | 0.291 | **0.937** | 0.934 | 0.924 | 0.934 |
| ColorReason | 0.950 | **0.950** | **0.950** | 0.400 | 0.400 | 0.393 | 0.660 | 0.515 | 0.930 | 0.940 | 0.935 | 0.810 |
| StratQA | 0.731 | 0.729 | 0.722 | 0.367 | 0.606 | 0.558 | 0.707 | 0.573 | **0.742** | 0.722 | 0.718 | 0.722 |
| TrackObj | 0.188 | 0.181 | 0.188 | 0.191 | 0.101 | 0.103 | 0.181 | 0.125 | 0.173 | 0.192 | 0.185 | **0.195** |
| UnitConv | 0.755 | **0.779** | 0.707 | 0.358 | 0.370 | 0.274 | 0.534 | 0.308 | 0.727 | 0.735 | 0.729 | 0.604 |
| VitaFact | 0.899 | **0.908** | 0.812 | 0.171 | 0.640 | 0.200 | 0.817 | 0.245 | 0.897 | 0.897 | 0.897 | 0.893 |
| WinoWhy | 0.802 | 0.797 | 0.767 | 0.002 | 0.028 | 0.038 | 0.005 | 0.344 | 0.797 | 0.767 | **0.801** | 0.795 |
| Average | 0.762 | 0.729 | 0.665 | 0.357 | 0.388 | 0.332 | 0.569 | 0.307 | 0.754 | 0.748 | **0.758** | 0.740 |

Table 5: Experimental results for tasks using accuracy as metric (LlaMA3-8B base model).

| Task Name | PEFT (reference) | LORA-F | LORA-B | Avg LoRA | TIES | DARE | Arrow | LoraHub | MeteoRA (T1-1k) | MeteoRA (T2-1k) | MeteoRA(T2-100) | MeteoRA(T2-5) |
|---|---|---|---|---|---|---|---|---|---|---|---|---|
| AbsNarr | 0.803 | **0.793** | 0.790 | 0.413 | 0.425 | 0.335 | 0.772 | 0.075 | 0.787 | 0.787 | 0.775 | 0.768 |
| ConParaKC | 0.999 | **0.999** | **0.999** | 0.514 | 0.594 | 0.492 | 0.997 | 0.219 | **0.999** | **0.999** | 0.976 | 0.992 |
| CSAlg | 0.841 | 0.841 | 0.841 | 0.705 | 0.686 | 0.663 | 0.780 | 0.602 | **0.845** | 0.826 | 0.826 | 0.830 |
| DisflQA | 0.703 | 0.680 | 0.605 | 0.374 | 0.396 | 0.377 | 0.504 | 0.197 | **0.706** | 0.703 | 0.686 | 0.628 |
| ElemMath | 0.780 | **0.777** | 0.606 | 0.273 | 0.308 | 0.245 | 0.645 | 0.106 | 0.776 | 0.773 | 0.751 | 0.725 |
| EpiReason | 1.000 | 0.996 | **1.000** | 0.430 | 0.450 | 0.425 | 0.600 | 0.170 | **1.000** | **1.000** | **1.000** | **1.000** |
| FormFall | 0.989 | 0.970 | 0.628 | 0.528 | 0.519 | 0.520 | 0.836 | 0.190 | **0.987** | **0.987** | 0.981 | 0.977 |
| GSWiki | 0.935 | 0.921 | 0.923 | 0.627 | 0.608 | 0.574 | 0.835 | 0.307 | **0.932** | 0.928 | 0.904 | 0.896 |
| GSM8K | 0.591 | 0.566 | 0.548 | 0.080 | 0.086 | 0.108 | 0.172 | 0.050 | 0.555 | **0.559** | 0.511 | 0.491 |
| LangID | 0.782 | 0.749 | 0.649 | 0.404 | 0.412 | 0.383 | 0.625 | 0.192 | **0.779** | 0.775 | 0.759 | 0.744 |
| LogDeduc | 0.760 | 0.707 | 0.707 | 0.403 | 0.423 | 0.383 | 0.627 | 0.367 | 0.757 | 0.753 | 0.747 | **0.770** |
| ObjCount | 0.880 | 0.555 | 0.865 | 0.060 | 0.080 | 0.130 | 0.005 | 0.230 | **0.875** | 0.850 | 0.785 | 0.750 |
| ParaSeg | 0.296 | 0.261 | 0.244 | 0.044 | 0.050 | 0.045 | 0.187 | 0.000 | **0.295** | 0.252 | 0.235 | 0.234 |
| PlayDiag | 0.649 | 0.632 | 0.649 | 0.647 | 0.650 | 0.644 | **0.656** | 0.092 | 0.649 | 0.649 | 0.580 | 0.581 |
| QuestSel | 0.936 | 0.911 | 0.930 | 0.544 | 0.506 | 0.472 | 0.845 | 0.247 | 0.927 | **0.940** | 0.892 | 0.892 |
| ColorReason | 0.958 | 0.945 | **0.965** | 0.565 | 0.595 | 0.530 | 0.793 | 0.238 | 0.960 | 0.983 | 0.915 | 0.905 |
| StratQA | 0.716 | 0.707 | **0.718** | 0.600 | 0.611 | 0.538 | 0.681 | 0.503 | 0.659 | 0.670 | 0.648 | 0.611 |
| TrackObj | 0.995 | 0.588 | 0.664 | 0.147 | 0.195 | 0.136 | 0.804 | 0.171 | 0.993 | **0.996** | 0.985 | 0.985 |
| UnitConv | 0.822 | 0.814 | 0.780 | 0.485 | 0.491 | 0.410 | 0.647 | 0.463 | 0.820 | 0.819 | 0.802 | 0.786 |
| VitaFact | 0.908 | 0.903 | 0.839 | 0.607 | 0.655 | 0.541 | 0.822 | 0.311 | **0.907** | **0.907** | 0.902 | 0.890 |
| WinoWhy | 0.816 | 0.797 | 0.802 | 0.524 | 0.516 | 0.526 | 0.750 | 0.203 | 0.818 | **0.827** | 0.788 | 0.788 |
| Average | 0.817 | 0.767 | 0.750 | 0.427 | 0.441 | 0.404 | 0.647 | 0.235 | **0.811** | 0.806 | 0.783 | 0.773 |

Table 6: Experimental results for tasks using BLEU and ROUGE as metrics (LlaMA2-13B base model).

| Task Name | Model | BLEU | ROUGE-1 | ROUGE-2 | ROUGE-L |
|---|---|---|---|---|---|
| | PEFT (reference) | 16.03 | 0.363 | 0.176 | 0.340 |
| ALPACA | LoRA-F | 23.96 | 0.302 | 0.140 | 0.283 |
| | LoRA-B | 11.72 | 0.341 | 0.157 | 0.317 |
| | Avg LoRA | 41.88 | 0.195 | 0.084 | 0.164 |
| | TIES | **80.34** | 0.209 | 0.092 | 0.175 |
| | DARE | 78.25 | 0.228 | 0.101 | 0.193 |
| | Arrow | 24.62 | 0.271 | 0.128 | 0.230 |
| | LoraHub | 0.00 | 0.240 | 0.117 | 0.206 |
| | MeteoRA (T1-1k) | 28.83 | **0.350** | **0.166** | **0.329** |
| | MeteoRA (T2-1k) | 24.12 | 0.349 | 0.162 | 0.327 |
| | MeteoRA (T2-100) | 39.09 | 0.332 | 0.160 | 0.281 |
| | MeteoRA (T2-5) | 12.49 | 0.306 | 0.140 | 0.256 |
| | PEFT (reference) | 7.50 | 0.228 | 0.067 | 0.214 |
| CNNDM | LoRA-F | 15.69 | 0.241 | **0.076** | **0.227** |
| | LoRA-B | 15.65 | 0.228 | 0.067 | 0.214 |
| | Avg LoRA | 13.08 | 0.144 | 0.032 | 0.104 |
| | TIES | 13.08 | 0.147 | 0.032 | 0.104 |
| | DARE | 13.08 | 0.126 | 0.031 | 0.081 |
| | Arrow | **17.42** | 0.173 | 0.043 | 0.122 |
| | LoraHub | 4.77 | 0.141 | 0.030 | 0.104 |
| | MeteoRA (T1-1k) | 7.50 | 0.229 | 0.069 | 0.216 |
| | MeteoRA (T2-1k) | 5.57 | 0.230 | 0.070 | 0.217 |
| | MeteoRA (T2-100) | 7.32 | 0.251 | 0.070 | 0.196 |
| | MeteoRA (T2-5) | 7.77 | **0.254** | 0.073 | 0.199 |
| | PEFT (reference) | 46.17 | 0.716 | 0.479 | 0.659 |
| LingPuzz | LoRA-F | 62.23 | 0.649 | 0.365 | 0.582 |
| | LoRA-B | 54.91 | 0.608 | 0.324 | 0.541 |
| | Avg Lora | 36.72 | 0.531 | 0.233 | 0.441 |
| | TIES | 49.14 | 0.405 | 0.117 | 0.308 |
| | DARE | **68.87** | 0.379 | 0.102 | 0.285 |
| | Arrow | 56.23 | 0.643 | 0.365 | 0.562 |
| | LoraHub | 0.00 | 0.172 | 0.057 | 0.131 |
| | MeteoRA (T1-1k) | 68.34 | 0.717 | 0.478 | **0.661** |
| | MeteoRA (T2-1k) | 46.17 | 0.713 | 0.476 | 0.655 |
| | MeteoRA (T2-100) | 46.17 | **0.718** | **0.480** | 0.646 |
| | MeteoRA (T2-5) | 57.47 | 0.716 | 0.474 | 0.646 |
| | PEFT (reference) | 78.25 | - | - | - |
| NewsDE | LoRA-F | 78.25 | - | - | - |
| | LoRA-B | 78.25 | - | - | - |
| | Avg Lora | 3.38 | - | - | - |
| | TIES | **86.48** | - | - | - |
| | DARE | **86.48** | - | - | - |
| | Arrow | **86.48** | - | - | - |
| | LoraHub | 50.09 | - | - | - |
| | MeteoRA (T1-1k) | **86.48** | - | - | - |
| | MeteoRA (T2-1k) | **86.48** | - | - | - |
| | MeteoRA (T2-100) | **86.48** | - | - | - |
| | MeteoRA (T2-5) | **86.48** | - | - | - |

| Task Name | Model | BLEU | ROUGE-1 | ROUGE-2 | ROUGE-L |
|-----------|-------|------|---------|---------|---------|
| | PEFT (reference) | 70.05 | - | - | - |
| | LoRA-F | 57.03 | - | - | - |
| | LoRA-B | **81.54** | - | - | - |
| | Avg Lora | 2.86 | - | - | - |
| | TIES | 70.05 | - | - | - |
| NewsES | DARE | 46.27 | - | - | - |
| | Arrow | **81.54** | - | - | - |
| | LoraHub | 0.64 | - | - | - |
| | MeteoRA (T1-1k) | **81.54** | - | - | - |
| | MeteoRA (T2-1k) | 70.05 | - | - | - |
| | MeteoRA (T2-100) | 70.05 | - | - | - |
| | MeteoRA (T2-5) | 70.05 | - | - | - |
| | PEFT (reference) | 39.04 | - | - | - |
| | LoRA-F | **54.90** | - | - | - |
| | LoRA-B | 40.08 | - | - | - |
| | Avg Lora | 40.20 | - | - | - |
| | TIES | 40.08 | - | - | - |
| NewsIT | DARE | 40.20 | - | - | - |
| | Arrow | 36.92 | - | - | - |
| | LoraHub | 40.08 | - | - | - |
| | MeteoRA (T1-1k) | 39.04 | - | - | - |
| | MeteoRA (T2-1k) | 39.04 | - | - | - |
| | MeteoRA (T2-100) | 39.04 | - | - | - |
| | MeteoRA (T2-1k) | 39.04 | - | - | - |
| | PEFT (reference) | 15.72 | 0.237 | 0.064 | 0.222 |
| | LoRA-F | 9.71 | 0.247 | **0.076** | **0.238** |
| | LoRA-B | 11.90 | **0.249** | 0.073 | 0.236 |
| | Avg Lora | 14.54 | 0.185 | 0.050 | 0.149 |
| | TIES | 16.62 | 0.112 | 0.024 | 0.088 |
| StratQA | DARE | 16.62 | 0.123 | 0.026 | 0.095 |
| | Arrow | 13.83 | 0.218 | 0.066 | 0.175 |
| | LoraHub | 11.50 | 0.171 | 0.038 | 0.128 |
| | MeteoRA (T1-1k) | 8.74 | 0.235 | 0.065 | 0.221 |
| | MeteoRA (T2-1k) | 10.03 | 0.240 | 0.068 | 0.226 |
| | MeteoRA (T2-100) | 13.95 | 0.222 | 0.063 | 0.172 |
| | MeteoRA (T2-5) | **20.69** | 0.228 | 0.067 | 0.174 |
| | PEFT (reference) | 12.50 | 0.157 | 0.027 | 0.146 |
| | LoRA-F | **28.39** | **0.153** | **0.025** | **0.142** |
| | LoRA-B | 9.78 | 0.143 | 0.021 | 0.134 |
| | Avg Lora | 1.21 | 0.099 | 0.010 | 0.060 |
| | TIES | 22.45 | 0.101 | 0.011 | 0.078 |
| TopChat | DARE | 22.48 | 0.103 | 0.011 | 0.065 |
| | Arrow | 11.16 | 0.099 | 0.013 | 0.080 |
| | LoraHub | 0.35 | 0.064 | 0.005 | 0.051 |
| | MeteoRA (T1-1k) | 13.44 | 0.151 | **0.025** | 0.141 |
| | MeteoRA (T2-1k) | 12.35 | 0.149 | **0.025** | 0.140 |
| | MeteoRA (T2-100) | 13.44 | 0.132 | 0.023 | 0.108 |
| | MeteoRA (T2-5) | 12.93 | 0.135 | 0.024 | 0.110 |

Table 7: Experimental results for tasks using BLEU and ROUGE as metrics (LlaMA3-8B base model).

| Task Name | Model | BLEU | ROUGE-1 | ROUGE-2 | ROUGE-L |
|---|---|---|---|---|---|
| | PEFT (reference) | 24.72 | 0.376 | 0.190 | 0.353 |
| | LoRA-F | 31.47 | 0.284 | 0.123 | 0.267 |
| | LoRA-B | 29.27 | **0.358** | **0.175** | **0.335** |
| | Avg LoRA | 73.49 | 0.206 | 0.089 | 0.172 |
| | TIES | 73.49 | 0.214 | 0.092 | 0.181 |
| ALPACA | DARE | 73.49 | 0.230 | 0.099 | 0.192 |
| | Arrow | 12.26 | 0.222 | 0.093 | 0.186 |
| | LoraHub | 0.00 | 0.176 | 0.068 | 0.151 |
| | MeteoRA (T1-1k) | 32.34 | **0.358** | 0.170 | **0.335** |
| | MeteoRA (T2-1k) | 30.08 | 0.354 | 0.170 | 0.332 |
| | MeteoRA (T2-100) | 31.19 | 0.317 | 0.147 | 0.266 |
| | MeteoRA (T2-5) | **80.34** | 0.249 | 0.103 | 0.204 |
| | PEFT (reference) | 11.93 | 0.231 | 0.069 | 0.218 |
| | LoRA-F | 16.13 | **0.248** | **0.080** | **0.233** |
| | LoRA-B | 13.27 | 0.233 | 0.070 | 0.218 |
| | Avg LoRA | 21.07 | 0.168 | 0.039 | 0.121 |
| | TIES | 18.07 | 0.154 | 0.037 | 0.109 |
| CNNDM | DARE | 4.67 | 0.137 | 0.032 | 0.096 |
| | Arrow | 13.13 | 0.153 | 0.037 | 0.111 |
| | LoraHub | 15.30 | 0.087 | 0.008 | 0.038 |
| | MeteoRA (T1-1k) | 11.93 | 0.233 | 0.070 | 0.218 |
| | MeteoRA (T2-1k) | 11.93 | 0.232 | 0.070 | 0.219 |
| | MeteoRA (T2-100) | **21.11** | 0.205 | 0.054 | 0.146 |
| | MeteoRA (T2-5) | 6.52 | 0.203 | 0.054 | 0.143 |
| | PEFT (reference) | 44.12 | 0.785 | 0.589 | 0.734 |
| | LoRA-F | 36.89 | 0.718 | 0.488 | 0.666 |
| | LoRA-B | 37.10 | **0.743** | **0.519** | **0.689** |
| | Avg LoRA | 28.87 | 0.421 | 0.134 | 0.331 |
| | TIES | 34.17 | 0.432 | 0.134 | 0.339 |
| LingPuzz | DARE | 56.23 | 0.357 | 0.113 | 0.281 |
| | Arrow | **59.00** | 0.721 | 0.505 | 0.659 |
| | LoraHub | 39.28 | 0.245 | 0.063 | 0.184 |
| | MeteoRA (T1-1k) | 41.72 | 0.695 | 0.451 | 0.636 |
| | MeteoRA (T2-1k) | 41.72 | 0.696 | 0.448 | 0.639 |
| | MeteoRA (T2-100) | 50.81 | 0.666 | 0.408 | 0.588 |
| | MeteoRA (T2-5) | 46.17 | 0.655 | 0.394 | 0.580 |
| | PEFT (reference) | 97.65 | - | - | - |
| | LoRA-F | 78.25 | - | - | - |
| | LoRA-B | 78.25 | - | - | - |
| | Avg LoRA | 63.56 | - | - | - |
| | TIES | 46.47 | - | - | - |
| NewsDE | DARE | 36.60 | - | - | - |
| | Arrow | 37.36 | - | - | - |
| | LoraHub | 11.87 | - | - | - |
| | MeteoRA (T1-1k) | **86.48** | - | - | - |
| | MeteoRA (T2-1k) | **86.48** | - | - | - |
| | MeteoRA (T2-100) | 51.42 | - | - | - |
| | MeteoRA (T2-5) | **86.48** | - | - | - |

| Task Name | Model | BLEU | ROUGE-1 | ROUGE-2 | ROUGE-L |
|---|---|---|---|---|---|
| | PEFT (reference) | 81.54 | - | - | - |
| | LoRA-F | **81.54** | - | - | - |
| | LoRA-B | **81.54** | - | - | - |
| | Avg LoRA | 31.18 | - | - | - |
| | TIES | 30.55 | - | - | - |
| NewsES | DARE | 17.61 | - | - | - |
| | Arrow | 31.82 | - | - | - |
| | LoraHub | 0.0 | - | - | - |
| | MeteoRA (T1-1k) | **81.54** | - | - | - |
| | MeteoRA (T2-1k) | **81.54** | - | - | - |
| | MeteoRA (T2-100) | **81.54** | - | - | - |
| | MeteoRA (T2-5) | 63.72 | - | - | - |
| | PEFT (reference) | 54.90 | - | - | - |
| | LoRA-F | **54.90** | - | - | - |
| | LoRA-B | 38.54 | - | - | - |
| | Avg LoRA | 38.54 | - | - | - |
| | TIES | 37.48 | - | - | - |
| NewsIT | DARE | 52.21 | - | - | - |
| | Arrow | 38.02 | - | - | - |
| | LoraHub | 0.0 | - | - | - |
| | MeteoRA (T1-1k) | **54.90** | - | - | - |
| | MeteoRA (T2-1k) | 51.83 | - | - | - |
| | MeteoRA (T2-100) | 35.22 | - | - | - |
| | MeteoRA (T2-5) | 36.78 | - | - | - |
| | PEFT (reference) | 10.58 | 0.249 | 0.077 | 0.236 |
| | LoRA-F | 10.44 | 0.234 | 0.068 | 0.223 |
| | LoRA-B | 10.58 | 0.243 | 0.071 | 0.230 |
| | Avg LoRA | **38.80** | 0.112 | 0.024 | 0.089 |
| | TIES | 10.90 | 0.102 | 0.022 | 0.082 |
| StratQA | DARE | 14.78 | 0.128 | 0.027 | 0.100 |
| | Arrow | 12.19 | 0.206 | 0.057 | 0.165 |
| | LoraHub | 14.35 | 0.147 | 0.033 | 0.116 |
| | MeteoRA (T1-1k) | 10.58 | **0.252** | 0.076 | **0.239** |
| | MeteoRA (T2-1k) | 10.58 | 0.250 | **0.077** | **0.239** |
| | MeteoRA (T2-100) | 20.56 | 0.228 | 0.065 | 0.174 |
| | MeteoRA (T2-5) | 11.67 | 0.213 | 0.055 | 0.162 |
| | PEFT (reference) | 39.50 | 0.151 | 0.025 | 0.141 |
| | LoRA-F | 33.82 | 0.150 | 0.024 | 0.140 |
| | LoRA-B | 19.22 | 0.139 | 0.019 | 0.131 |
| | Avg LoRA | 23.59 | 0.094 | 0.012 | 0.078 |
| | TIES | 26.13 | 0.092 | 0.011 | 0.077 |
| TopChat | DARE | 38.31 | 0.086 | 0.008 | 0.066 |
| | Arrow | 35.64 | 0.112 | 0.016 | 0.091 |
| | LoraHub | 0.08 | 0.049 | 0.002 | 0.031 |
| | MeteoRA (T1-1k) | **45.64** | **0.152** | **0.026** | **0.141** |
| | MeteoRA (T2-1k) | **45.64** | **0.152** | 0.024 | **0.141** |
| | MeteoRA (T2-100) | 27.36 | 0.129 | 0.021 | 0.107 |
| | MeteoRA (T2-5) | 40.86 | 0.130 | 0.018 | 0.109 |

### A.5  *Composite-n* EVALUATION RESULTS DETAILS

The task construction method for the *composite-n* series is similar across different sets. Taking *composite-10* as an example, each sample in this test set can be thought of as a "test sheet" containing 10 questions presented in sequence. During evaluation, this test sheet is provided as input to the LLM and ask it to output the answers along with the corresponding question numbers in order. To ensure that the model is capable of answering these 10 questions, we select 10 tasks from the 28 selected tasks, ensuring diversity in knowledge domains and question formats. Each sample in the *composite-10* task is constructed by randomly sampling one instance from each of the 10 tasks (without repetition) and concatenating them in sequence. However, given the limited capability of the instruction following in the zero-shot setting, neither the MeteoRA models nor the models fine-tuned by LoRA achieve satisfactory results. Hence, we employ a 2-shot setting for evaluation on these *composite-n* tasks.

The evaluation metrics used for *composite-n* tasks are: average number of questions attempted, average number of multiple-choice questions answered correctly, and average BLEU, ROUGE scores for non-multiple-choice questions.

Notice that in the *composite-n tasks*, when calculating the softmax values of the weights for the two LoRA adapters selected by the Gating network, we introduced a hyperparameter called *temperature*. The value of *temperature* needs to be increased as the number of sub-tasks grows. Specifically, we set the *temperature* values to 15, 20, and 30 for the three tasks, respectively.

Tables 8, 9, and 10 present the detailed evaluation results for the *composite-3*, *composite-5*, and *composite-10* tasks, respectively. Several important clarifications are necessary for interpreting these results:

1. The models are required to generate both the corresponding question number and its answer. Any mismatch between the question number and the answer is therefore considered incorrect.

2. In the evaluation results, some BLEU scores are recorded as 0. This occurs when the model generates mismatched question numbers and answers or provides extremely insufficient answers, resulting in an overall 0 BLEU score.

3. For the task *strategyqa*, which involves answering with either 'yes' or 'no' and providing reasoning steps, the accuracy metric specifically measures the correctness of the 'yes' or 'no' response.

4. The reported ROUGE scores refer to the F1-scores.

5. Samples that the lengths exceed to 4,096 tokens are skipped in the evaluation process (we skip 13 samples in total).

Table 8: The *composite-3* evaluation results are presented in details with MeteoRA results on the left side and LoRA-B results on the right side of each metric column. A dash ('-') indicates that the corresponding metric was not applicable or included in the evaluation.

| Sub-task Name | Accuracy↑ | | BLEU↑ | | ROUGE-1↑ | | ROUGE-2↑ | | ROUGE-L↑ | |
|---|---|---|---|---|---|---|---|---|---|---|
| LogDeduc | 0.500↑ | 0.430 | - | - | - | - | - | - | - | - |
| QuestSel | 0.545↓ | 0.630 | - | - | - | - | - | - | - | - |
| StratQA | 0.445↑ | 0.250 | 15.31 | 10.55 | 0.195↑ | 0.135 | 0.052↑ | 0.027 | 0.182↑ | 0.128 |

Table 9: The *composite-5* evaluation results are presented in details with MeteoRA results on the left side and LoRA-B results on the right side of each metric column. A dash ('-') indicates that the corresponding metric was not applicable or included in the evaluation.

| Sub-task Name | Accuracy↑ | | BLEU↑ | | ROUGE-1↑ | | ROUGE-2↑ | | ROUGE-L↑ | |
|---|---|---|---|---|---|---|---|---|---|---|
| LogDeduc | 0.500 | 0.500 | - | - | - | - | - | - | - | - |
| QuestSel | 0.620↓ | 0.770 | - | - | - | - | - | - | - | - |
| AbsNarr | 0.350↓ | 0.460 | - | - | - | - | - | - | - | - |
| GSWiki | 0.650↑ | 0.410 | - | - | - | - | - | - | - | - |
| StratQA | 0.495↑ | 0.275 | 9.86↑ | 9.41 | 0.221↑ | 0.219 | 0.069↑ | 0.063 | 0.207↓ | 0.208 |

Table 10: The *composite-10* evaluation results are presented in details with MeteoRA results on the left side and LoRA-B results on the right side of each metric column. A dash ('-') indicates that the corresponding metric was not applicable or included in the evaluation. Note that the 0.00 BLEU scores are caused by mismatch and too insufficient answers.

| Sub-task Name | Accuracy↑ | | BLEU↑ | | ROUGE-1↑ | | ROUGE-2↑ | | ROUGE-L↑ | |
|---|---|---|---|---|---|---|---|---|---|---|
| LogDeduc | 0.500↑ | 0.453 | - | - | - | - | - | - | - | - |
| QuestSel | 0.703↑ | 0.688 | - | - | - | - | - | - | - | - |
| AbsNarr | 0.625↓ | 0.672 | - | - | - | - | - | - | - | - |
| GSWiki | 0.773↑ | 0.727 | - | - | - | - | - | - | - | - |
| WinoWhy | 0.422↑ | 0.078 | - | - | - | - | - | - | - | - |
| StratQA | 0.461↑ | 0.211 | 3.23↑ | 0.00 | 0.225↑ | 0.106 | 0.051↑ | 0.025 | 0.210↑ | 0.099 |
| DisflQA | 0.266↑ | 0.117 | - | - | - | - | - | - | - | - |
| NewsDE | - | - | 14.78↑ | 14.54 | - | - | - | - | - | - |
| ALPACA | - | - | 0.00↓ | 8.17 | 0.257↑ | 0.187 | 0.075 | 0.075 | 0.241↑ | 0.167 |
| LingPuzz | - | - | 17.37↑ | 12.14 | 0.233↑ | 0.189 | 0.052↑ | 0.030 | 0.176↑ | 0.103 |

### A.6 FULL CONTENT OF THE ILLUSTARTED *composite-n* TASK

Below is the full content of a *composite-3* task, which includes 2-shot examples, the input (3 problems), and the model generated output, shown in Figure 4.

Example 1:\n\n[INST] The following paragraphs each describe a set of five objects arranged in a fixed order. The statements are logically consistent within each paragraph.\n\nOn a branch, there are five birds: a cardinal, a crow, a falcon, a robin, and a blue jay. The robin is to the right of the cardinal. The cardinal is to the right of the blue jay. The blue jay is the second from the left. The crow is the rightmost. Which choice is correct?\nchoice: The cardinal is the leftmost.\nchoice: The crow is the leftmost.\nchoice: The falcon is the leftmost.\nchoice: The robin is the leftmost.\nchoice: The blue jay is the leftmost.\n[/INST]\n\n[INST] Given a short answer along with its context, select the most appropriate question which has the given short answer as its answer.\n\nHere is the short answer followed by the context.\nShort Answer: magazines and journals Context: Tesla wrote a number of books and articles for magazines and journals. Among his books are My Inventions: The Autobiography of Nikola Tesla, compiled and edited by Ben Johnston; The Fantastic Inventions of Nikola Tesla, compiled and edited by David Hatcher Childress; and The Tesla Papers.\nchoice: Who was the first to post tesla's writings?\nchoice: Who was in charge of editing tesla's autobiography?\nChoose the appropriate question which has the given answer. [/INST]\n\n[INST] Q: Can Hulk's alter ego explain atomic events?\nA: [/INST]\n\n1. The falcon is the leftmost.\n2. Who was the first to post tesla's writings?\n3. Yes. Hulk's alter ego is Dr. Robert Bruce Banner. Dr. Robert Bruce Banner is a nuclear physicist. Nuclear physics is the field of physics that studies atomic nuclei and their constituents and interactions.\n\n\nExample 2:\n\n[INST] The following paragraphs each describe a set of three objects arranged in a fixed order. The statements are logically consistent within each paragraph.\n\nIn a golf tournament, there were three golfers: Ana, Rob, and Joe. Joe finished above Ana. Rob finished above Joe. Which choice is correct?\nchoice: Ana finished second.\nchoice: Rob finished second.\nchoice: Joe finished second.\n[/INST]\n\n[INST] Given a short answer along with its context, select the most appropriate question which has the given short answer as its answer.\n\nHere is the short answer followed by the context.\nShort Answer: two months Context: It was not until January 1518 that

friends of Luther translated the 95 Theses from Latin into German and printed and widely copied them, making the controversy one of the first in history to be aided by the printing press. Within two weeks, copies of the theses had spread throughout Germany; within two months, they had spread throughout Europe.\nchoice: How long did the theses take to spread through europe?\n choice: How long did it take for the printing of the theses to spread through germany?\nChoose the appropriate question which has the given answer. [/INST]\n\n[INST] Q: Can Iowa be hidden in the English Channel?\nA: [/INST]\n\n1. Joe finished second.\n2. How long did the theses take to spread through europe?\n3. No. The maximum width of the English Channel is 150 miles. The minimum width of Iowa is 200 miles.\n\n\nFollowing the example above, generate answers to the questions below:\n\n[INST] The following paragraphs each describe a set of seven objects arranged in a fixed order. The statements are logically consistent within each paragraph.\n\nOn a branch, there are seven birds: a cardinal, a blue jay, a robin, a falcon, a quail, a hummingbird, and a raven. The falcon is to the right of the quail. The robin is to the right of the hummingbird. The raven is to the left of the quail. The blue jay is the second from the left. The robin is the third from the left. The cardinal is the third from the right. Which choice is correct?\nchoice: The cardinal is the second from the right.\nchoice: The blue jay is the second from the right.\nchoice: The robin is the second from the right.\nchoice: The falcon is the second from the right.\nchoice: The quail is the second from the right.\nchoice: The hummingbird is the second from the right.\nchoice: The raven is the second from the right.\n [/INST]\n\n[INST] Given a short answer along with its context, select the most appropriate question which has the given short answer as its answer.\n\nHere is the short answer followed by the context.\nShort Answer: Ghazan Khan Context: The invasions of Baghdad, Samarkand, Urgench, Kiev, Vladimir among others caused mass murders, such as when portions of southern Khuzestan were completely destroyed. His descendant Hulagu Khan destroyed much of Iran's northern part and sacked Baghdad although his forces were halted by the Mamluks of Egypt, but Hulagu's descendant Ghazan Khan would return to beat the Egyptian Mamluks right out of Levant, Palestine and even Gaza. According to the works of the Persian historian Rashid-al-Din Hamadani, the Mongols killed more than 70,000 people in Merv and more than 190,000 in Nishapur. In 1237 Batu Khan, a grandson of Genghis Khan, launched an invasion into Kievan Rus'. Over the course of three years, the Mongols destroyed and annihilated all of the major cities of Eastern Europe with the exceptions of Novgorod and Pskov.\n choice: Which genghis khan descendant sacked baghdad?\n choice: Which of eastern europe's big cities survived the mongol invasion?\n choice: Which of genghis khan's descendants was responsible for driving the mamluks from palestine?\nChoose the appropriate question which has the given answer. [/INST]\n\n[INST] Q: Could the main character of "Alice's Adventures in Wonderland" join a Masonic Lodge?\nA: [/INST]\n\n1. The quail is the second from the right.\n2. Which of genghis khan's descendants was responsible for driving the mamluks from palestine?\n3. No. The main character of "Alice's Adventures in Wonderland" is Alice. Women are not allowed to join Masonic Lodges.

| | logical_deduction | | question_selection | | strategyqa | | other task |

### A.7    EFFICIENCY EVALUATION EXPERIMENTS ON DIFFERENT METEORA FORWARD PASS IMPLEMENTATIONS

In addition to experiments on our 28 selected tasks, we assess the efficiency of our MeteoRA forward pass design using randomly-generated pseudo data across various settings, including batch size ($b$), sequence length ($s$), gating weights top-k ($k$), LoRA rank size ($r$), number of LoRAs ($l$), maximum tokens to generate ($g$), input hidden dimension ($h$), and output hidden dimension ($hout$). Moreover, here we introduce a new baseline, *loop-speedup*, which improves upon *loop-original* by removing redundant or inefficient operations directly, acting like a strong substitute for the original design.

As depicted in Figures 7 for memory efficiency and 8 for time efficiency, our *bmm-torch* design outperforms other implementations, boasting an average speedup of $\sim 4\times$ over *loop-original*. However, its memory usage escalates with longer sequence lengths. In contrast, *bmm-triton* maintains a comparable memory footprint to the baselines while retaining $80\%$ of the speedup achieved by *bmm-torch*, showcasing a balanced trade-off between time and space, as illustrated in Figure 6 for overall efficiency.

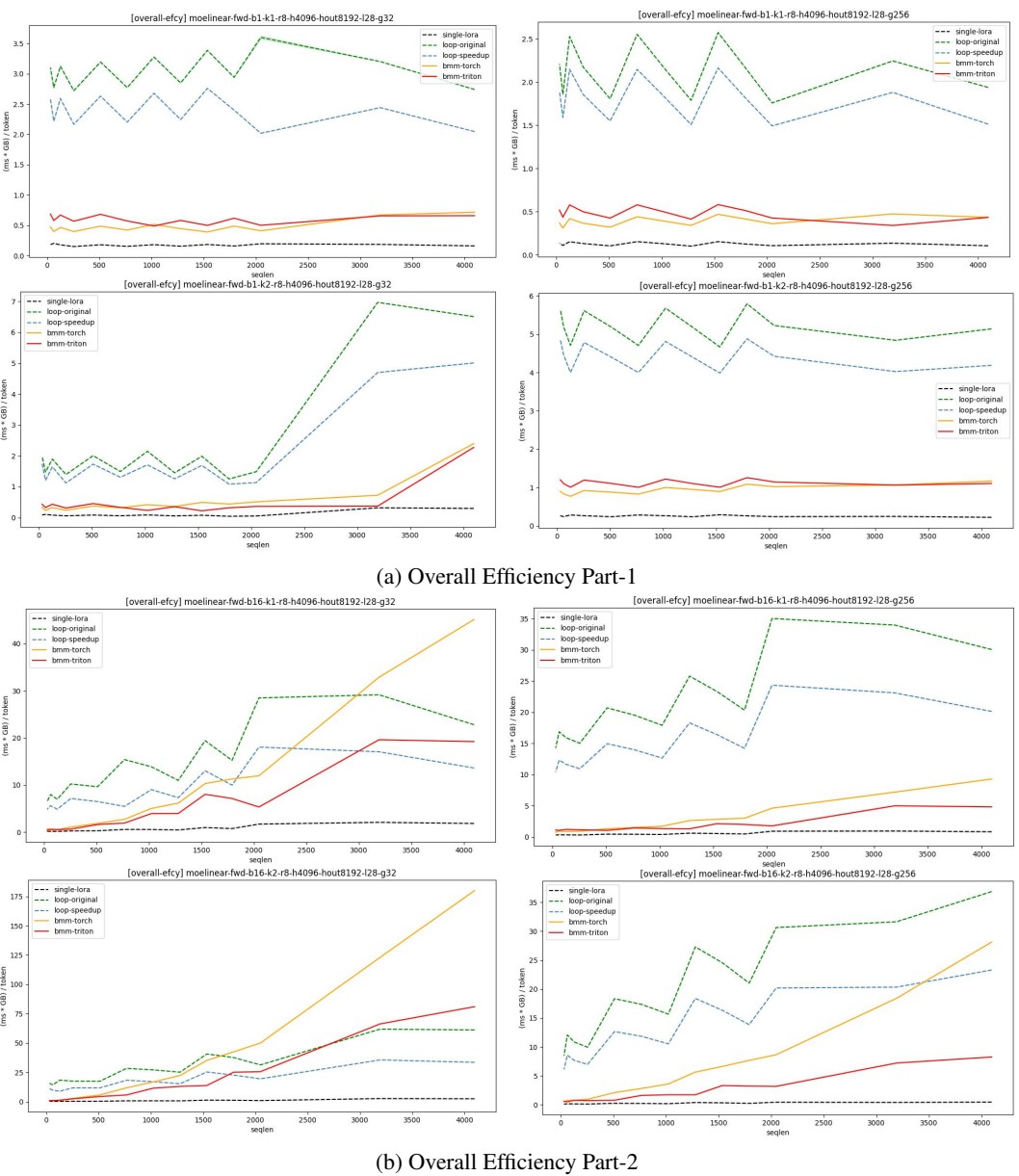

Figure 6: The overall efficiency evaluation curve displays the averaging runtime × memory footprint for each newly generated token (unit: ms × GB / token).

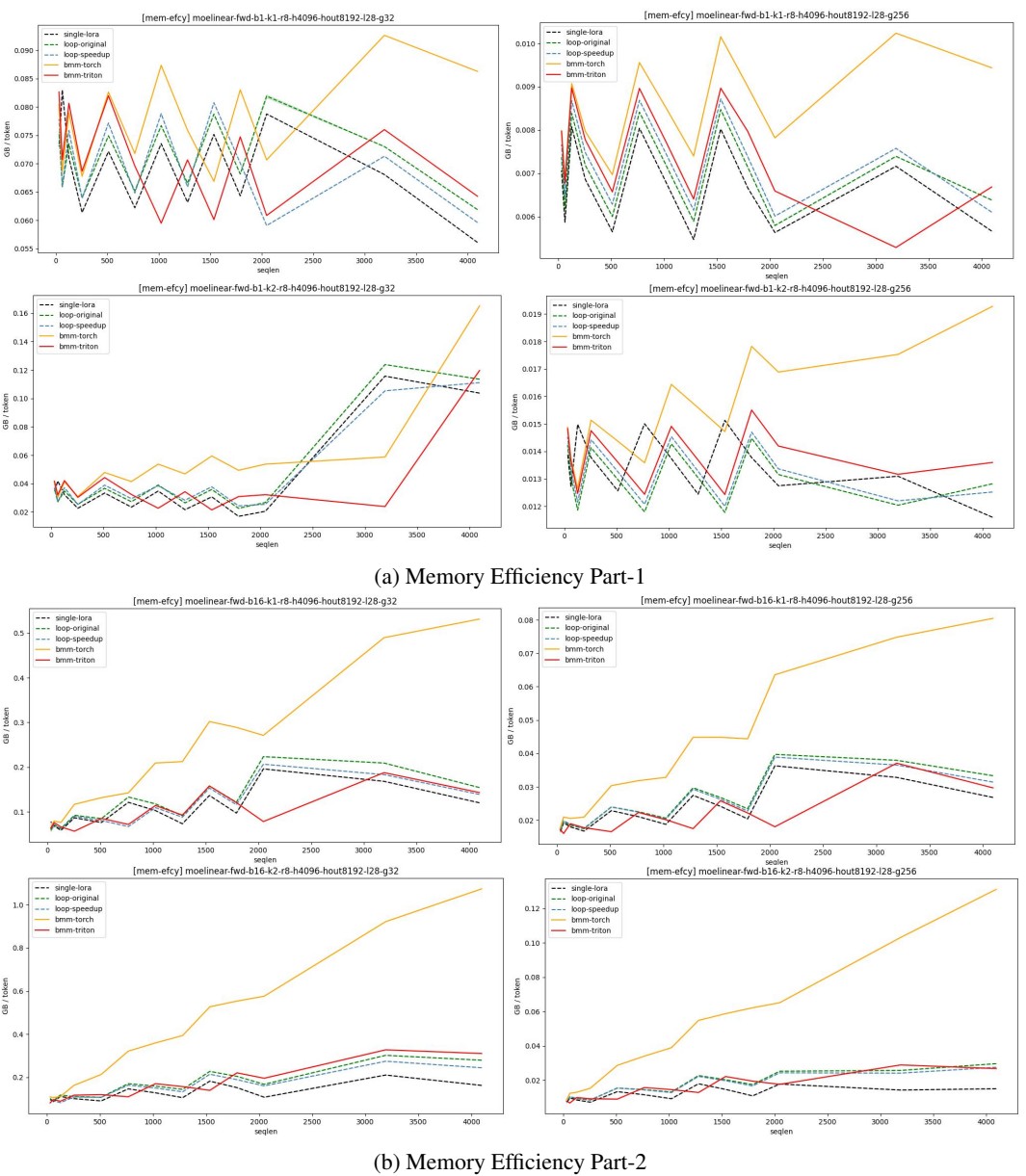

Figure 7: The memory efficiency evaluation curve displays the averaging memory footprint for each newly generated token (unit: GB / token).

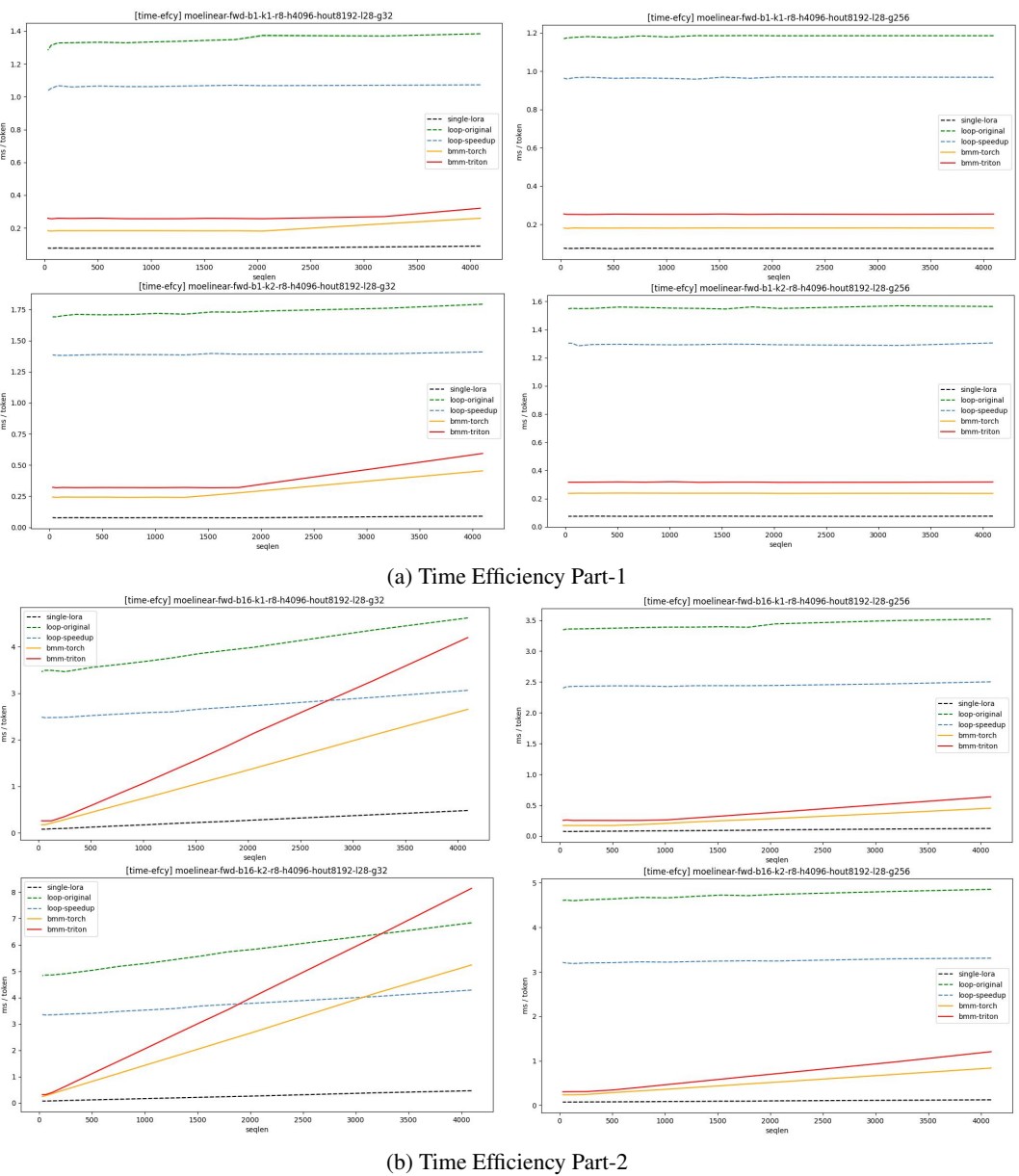

Figure 8: The time efficiency evaluation curve displays the averaging runtime for each newly generated token (unit: ms / token).

