# OpenReview forum: "MeteoRA: Multiple-tasks Embedded LoRA for Large Language Models"
_ICLR.cc/2025/Conference — ICLR 2025 Poster_

### Official Review · Reviewer_4Xew · 2024-10-30

**Soundness:** 3
**Presentation:** 3
**Contribution:** 2
**Rating:** 6
**Confidence:** 4

**Summary:**

The authors propose an automatic selection model for multi-task LoRA using a MoE arch, supporting both top-1 and top-k selection methods. In addition to constructing an automatic selection process using Gate Logits, they also utilize PyTorch's bmm operator for forward acceleration. The authors demonstrate the effectiveness of their proposed algorithm across multiple tasks by comparing it with existing multi-task LoRA methods as well as their own baselines, LoRA-F and LoRA-B.

**Strengths:**

### Originality
Since the gate logits approach is primarily inspired by methods from Mixtral of Experts, the main novelty lies in the design of the loss function and the implementation of forward acceleration. This method can solve up to ten sequential problems in a single inference pass automatically, which demonstrates both the scalability and utility of the proposed solution.

### Quality
The technical details appear correct, as the paper does not involve extensive mathematical derivations. The authors also introduce efficient acceleration strategies for MoE, which contribute meaningfully to improving the computational feasibility of such models.

### Clarity
The paper is generally clear in presenting the proposed methodology and results.

### Significance
The work addresses a significant challenge in the field of parameter-efficient fine-tuning (PEFT) by proposing a novel approach to autonomously manage and switch between multiple LoRA adapters embedded within a single LLM. The proposed framework is of considerable practical value for large-scale language models and downstream applications. The emphasis on practical deployment of the framework and the specific use of LoRA in a multi-task setting distinguishes this work from previous studies.

**Weaknesses:**

### Originality
The gate logits approach is primarily inspired by methods from Mixtral of Experts, and the paper could more clearly emphasize its origin and what it is (just a linear layer with softmax).

### Clarity
The description of the gating network is insufficient. More details on the construction of composite-3/5/10 tasks should also be provided in the main text. More motivations should be included.

### Writing Quality
The LaTeX formatting should be unified, such as using consistent notation for $o_{base}$ and $W_\mathrm{base}$ in Equation (2). Additionally, consistent notation should be used in the formulas (e.g., whether the vectors are row or column vectors, and whether matrices operate on vectors from the left or the right). For instance, in Equation (2), the expression $o = xW_\mathrm{base} + x\Delta W_{I(x)}$ is used, while in Appendix A, $h = W_\mathrm{base}x + B_iA_ix$ is used.

**Questions:**

1. In the appendix, the authors mention, "However, given the limited capability of the instruction following in the zero-shot setting, neither the MeteoRA models nor the models fine-tuned by LoRA achieve satisfactory results." Is there any supporting evidence for this statement? Also, why did the authors ultimately choose a 2-shot setting?
2. According to Figure 3, MeteoRA's performance on the ParaSeg task is noticeably poor when using LlaMA2, but it shows significant improvement with LlaMA3. Could the authors provide an explanation for this?

---

> ### Author Response · Authors · 2024-11-22
> **Response to weaknesses related to originality, clarity, and writing quality**
>
> ## W1: Originality
>
> Thank you for your suggestion. We will revise the paper to include this explanation and add the corresponding references.
>
> ## W2: Clarity - Description of the Gating Network
>
> Thank you for your suggestion. We will revise the paper to include a more detailed explanation of the Gating Network. Additionally, we will update Figure 2 to improve clarity and make it easier to understand.
>
> ## W2: Clarity - Details on the Construction of Composite Tasks
>
> Thank you for your suggestion. The construction method for the *composite-3*, *composite-5*, and *composite-10* tasks is consistent, with the only difference being the number of questions in each sample. Taking the *composite-10* task as an example, you can imagine a test sheet of an exam containing 10 sequential questions. We provide this "test sheet" as input to the LLM and ask it to output the answers along with the corresponding question numbers in order.
>
> To ensure that the model is capable of solving all 10 questions, we select them from 28 selected tasks, ensuring diversity in knowledge domains and question formats. Each sample in the *composite-10* task is constructed by sampling one question (non-repeated) from the test set of each of the 10 selected tasks and concatenating them sequentially.
>
> For simplicity, here is an example of the format for a *composite-3* task, along with the expected correct output:
>
> ```
> input:
> [INST]
> The following paragraphs each describe a set of five objects arranged in a fixed order. The statements are logically consistent within each paragraph.
>
> On a branch, there are five birds: a cardinal, a crow, a falcon, a robin, and a blue jay. The robin is to the right of the cardinal. The cardinal is to the right of the blue jay. The blue jay is the second from the left. The crow is the rightmost. Which choice is correct?\nchoice: The cardinal is the leftmost.
> choice: The crow is the leftmost.
> choice: The falcon is the leftmost.
> choice: The robin is the leftmost.
> choice: The blue jay is the leftmost.
> [/INST]
>
> [INST]
> Given a short answer along with its context, select the most appropriate question which has the given short answer as its answer.
>
> Here is the short answer followed by the context.
> Short Answer: magazines and journals Context: Tesla wrote a number of books and articles for magazines and journals. Among his books are My Inventions: The Autobiography of Nikola Tesla, compiled and edited by Ben Johnston; The Fantastic Inventions of Nikola Tesla, compiled and edited by David Hatcher Childress; and The Tesla Papers.
> choice: Who was the first to post tesla's writings?
> choice: Who was in charge of editing tesla's autobiography?
> Choose the appropriate question which has the given answer.
> [/INST]
>
> [INST]
> Q: Can Hulk's alter ego explain atomic events?
> A:
> [/INST]
>
> output:
> 1. The falcon is the leftmost.
> 2. Who was the first to post tesla's writings?
> 3. Yes. Hulk's alter ego is Dr. Robert Bruce Banner. Dr. Robert Bruce Banner is a nuclear physicist.  Nuclear physics is the field of physics that studies atomic nuclei and their constituents and interactions.
>
> ```
>
> We will include this explanation in the appendix of the paper and provide examples to ensure clarity and ease of understanding.
>
> ## W3: Writing Quality
>
> Thank you for your suggestion. We will revise the formulas in the paper to ensure consistency between the main text and the appendix.

---

> ### Author Response · Authors · 2024-11-22
> **Response to questions 1 and question 2**
>
> ## Q1: Clarity of the Experiment Setting
>
> Thank you for your question. The statement reflects the results of our experiments. All the baselines were trained on samples formatted as $\langle \text{question}, \text{answer} \rangle$, without exposure to concatenated samples in the form of $\langle\text{concat(question}\_1, \text{question}\_2, \ldots, \text{question}\_n), \text{concat(answer}\_1, \ldots, \text{answer}\_n) \rangle$. As a result, most baselines fail to perform satisfactorily on *composite-n* tasks. Specifically, they tend to either answer only $\text{question}_1$ or $\text{question}_n$, ignoring the other questions entirely.
>
> The choice to use a 2-shot setting was based on two considerations:
>
> 1. **Output Format Alignment**: To facilitate evaluation, we required the model to output both the question index and the corresponding answer. This specific output format had not been learned by the models during training. Providing examples was necessary to guide the models toward producing outputs that adhered to this requirement.
>
> 2. **Minimal Effective Prompting**: While we initially tested a 1-shot setting, none of the models achieved satisfactory performance under this condition. The 2-shot setting emerged as the minimal configuration capable of enabling the models to handle *composite-n* tasks effectively.
>
> ## Q2: Explanation for Performance on the *ParaSeg* task
>
> Thank you for your question. We would like to provide further clarification on these two issues. In MeteoRA, the performance drop on certain tasks when $k=2$ or even $k=1$ arises from a combination of factors. Based on our experimental setup, the following influences are noteworthy:
>
> 1. **Base Model Capability**: The overall performance of MeteoRA on LLaMA3-8B is slightly better than on LLaMA2-13B, indicating that the base model's capacity affects the results.
>
> 2. **Task Selection**: When selecting tasks, we prioritized diversity in task formats and domains while intentionally including some tasks with overlapping characteristics to observe potential interactions. With 28 tasks, conflicts between the corresponding LoRA adapters are inevitable, which can lead to performance degradation.
>
> 3. **LoRA Adapter Quality**: Since the output of one LoRA adapter affects the subsequent Gating Network’s choices, the quality of individual LoRA adapters can impact MeteoRA’s overall performance.
>
> 4. **Training Approach**: The LoRA-F baseline combines the full training datasets of all tasks into a single concatenated dataset. This method inherently prioritizes tasks with larger datasets (e.g., CNNDM) while significantly compromising performance on tasks with smaller datasets (e.g., ObjCount). In contrast, our approach fine-tuned MeteoRA using a maximum of 1,000 samples per task. Consequently, while MeteoRA may exhibit slight underperformance on certain tasks that require larger datasets to fine-tune the gating mechanism effectively, this trade-off aligns with our focus on balanced and efficient task performance.
>
> Additionally, MeteoRA's primary trade-off is ensuring that all tasks maintain reasonably consistent performance without significant degradation. While performance on a small subset of tasks may slightly lag behind the baseline, the overall gap between MeteoRA and PEFT is minimal, and both methods achieve satisfactory results across tasks.

---

> ### Comment · Reviewer_4Xew · 2024-11-26
>
> Thanks for the response. I will raise the score to 6.

---

### Official Review · Reviewer_4FYx · 2024-11-01

**Soundness:** 2
**Presentation:** 3
**Contribution:** 2
**Rating:** 6
**Confidence:** 2

**Summary:**

This paper proposes MeteoRA to enable scalable multi-task LoRA embedding within LLMs. The key of MeteoRA is using gated MoE to automatically select the most pertinent LoRA adapters to generate appropriate responses. It also employs efficient GPU kernel operators for forward acceleration. Evaluation results show the effectiveness of MeteoRA.

**Strengths:**

+ Using gated MoE for automatic selection of LoRA adapters
+ Employing efficient GPU kernel operators for forward acceleration

**Weaknesses:**

- Re-training is required when some LoRA adapters are updated, making it hard to use in practice

**Questions:**

How to reduce the re-training costs when some LoRA adapters are updated? How many LoRA adapters can be supported by MeteoRA?

---

> ### Author Response · Authors · 2024-11-22
> **Response to Reviewer 4FYx**
>
> ## W1: Practicality Issue: Re-training Required for Updated LoRA Adapters
>
> Thank you for your question. In our paper, we included three baseline methods that do not require re-training when LoRA adapters are updated: TIES, DARE, and Arrow, with the first two supported by HuggingFace's PEFT library. Here is a brief comparison of these methods:
>
> * **TIES and DARE**: These methods aim to merge models fine-tuned for different downstream tasks. While HuggingFace’s PEFT library provides support for these strategies, their performance significantly deteriorates when merging LoRA adapters across 28 tasks. Furthermore, these methods require manual adjustment of the weights for each LoRA adapter, meaning they cannot automatically switch to the appropriate LoRA based on user input.
>
> * **Arrow**: This method uses a routing algorithm to reuse LoRA adapters, leveraging the right first singular vector derived from the SVD of the product $A\_iB\_i^T$ for each LoRA adapter.
>
> As shown in Table 1 of our paper, TIES and DARE exhibit significant performance drops if the weights of each LoRA adapter are not carefully tuned. Similarly, Arrow struggles to achieve satisfactory performance at the scale of 28 tasks.
>
> In our point of view, learning a model once to marge arbitrary set of LoRA adapters is not applicable. Given an arbitrary set of LoRA adapters (sampled from an universal set of LoRA adapters), and each LoRA adapter is tuned on a task-specific distribution. Assume a learner, A, can learn the optimal weight for LoRA Merge.  Since different tasks have different data distributions, the corresponding LoRA adapters are fitted to different distributions. If the optimal weight in learner A exists, learner A would be a "universal learner." However, the No Free Lunch Theorem tells us that no such learner can exist. This means that learner A cannot learn a set of weights to merge all different LoRA adapters to perform well across all inference tasks.
>
> ## Q1: Reducing Re-training Costs for Updated LoRA Adapters
>
> Thank you for your question. This issue is the same as the one raised in W1. Please refer to our response to W1 for further clarification.
>
> Although "No Free Lunch Theorem" shows impractical of directly mixing LoRA adapters, there are still some potential ways to reducing the re-training costs when updating LoRA adapters. If each LoRA adapter is published with a standard manual (the manual template should be defined beforehand) related to its training set distribution, one may construct a complex distribution with some statistical approaches when applying multiple LoRA adapters together. The statistical approaches are usually more lightweight compared to fine-tuning parameters inside LLM.
>
> ## Q2: Capacity of MeteoRA: Number of Supported LoRA Adapters
>
> Thank you for your question. The capacity of MeteoRA can be examined from two perspectives:
>
> 1. **Memory Cost:** One aspect of capacity relates to memory requirements. In our experiments, all LoRA weights combined account for less than 10% of the base model weights (e.g., \~7% for 28 LoRA experts), resulting in a memory cost of approximately 1.9 GB for LLaMA3-8B equipped with MeteoRA in BF16 format. For instance, loading 100 LoRA adapters simultaneously would require less than 8 GB of memory. Additionally, since MeteoRA employs an MoE-style gating mechanism, the memory cost associated with activations remains constant, regardless of the number of loaded LoRA adapters.
>
> 2. **Gating Network Performance:** Another critical capacity metric is the performance of the gating network in correctly classifying the appropriate LoRA adapter. In our evaluation, MeteoRA embeds 28 LoRA adapters into a single base model . In contrast, prior work \[1, 2, 3] and others typically embed only up to 8 LoRA adapters/tasks (i.e., datasets). Some of these methods require explicit task or adapter identities as input parameters during evaluation, lacking the autonomous sensing and switching capabilities provided by MeteoRA.
>
> Due to time constraints, we regret that we could not conduct further evaluations on the gating network’s capacity, such as scaling MeteoRA to support 50 LoRA adapters. However, from our observations, MeteoRA with 28 LoRA adapters has demonstrated robustness. LLaMA3-8B with MeteoRA includes 224 gating networks (7 gating networks per decoder layer across 32 decoder layers in LLaMA3-8B). Even if some gating networks misclassify the appropriate LoRA adapter, the remaining networks can rectify the errors and ensure proper generation during LLM inference.
>
> \[1] When MOE Meets LLMs: Parameter Efficient Fine-tuning for Multi-task Medical Applications, SIGIR 2024.
>
> \[2] Li, Dengchun, et al. "Mixlora: Enhancing large language models fine-tuning with lora based mixture of experts." *arXiv preprint arXiv:2404.15159* (2024).
>
> \[3] Feng, Wenfeng, et al. "Mixture-of-loras: An efficient multitask tuning for large language models." *arXiv preprint arXiv:2403.03432* (2024).

---

> > ### Comment · Reviewer_4FYx · 2024-11-25
> >
> > Thanks for the response. I keep my positive score.

---

### Official Review · Reviewer_dk7r · 2024-11-01

**Soundness:** 3
**Presentation:** 3
**Contribution:** 3
**Rating:** 6
**Confidence:** 3

**Summary:**

The paper introduces MeteoRA, a framework designed to enhance the deployment of multiple LoRA adapters in LLM through a Mixture-of-Experts (MoE) architecture. This approach aims to facilitate autonomous task sensing and dynamic adapter switching, improving efficiency in handling composite tasks.

**Strengths:**

- The use of a full-mode MoE architecture to integrate multiple LoRA adapters is a novel contribution, potentially addressing limitations in existing methods like Huggingface PEFT and S-LoRA.
- The proposed forward acceleration strategies address efficiency challenges in traditional MoE implementations, achieving significant speedups.

**Weaknesses:**

- It will be better to also compare with a model trained with MoE upcycling and discuss the benefit of the proposed method.
- It should be a more detailed analysis of the triton operator, how it differ from methods like S-LoRA.
- The legend in Figure 3 is too small

**Questions:**

See weakness

---

> ### Author Response · Authors · 2024-11-22
> **Response to Reviewer dk7r**
>
> ## W1: Comparison with MoE Upcycling
>
> Thank you for your question. We would like to clarify the differences between MoE upcycling and MeteoRA in detail.
>
> The first key distinction lies in their usage scenarios:
>
> * **MoE Upcycling** is a method for extending dense models into MoE (Mixture of Experts) models, primarily aimed at expanding model parameters to achieve better overall capabilities. In this approach, the FFN layers are duplicated across the number of experts, initializing each expert’s weights to ensure stability during training. All experts are trainable during this process.
>
> * **MeteoRA**, on the other hand, is designed to reuse multiple existing LoRA adapters to deal with multitask scenarios, enabling autonomous selection and automatic switching of LoRA adapters based on user input. It aims to integrate a set of pre-existing LoRA adapters (either user-provided or publicly available) into a single model, since each existing LoRA adapter is fine-tuned for a specific application domain/task. So we could leverage numerous existing LoRA adapters from the market (e.g., HuggingFace). With a very small set of examples (5 to 100 in our evaluation) along with the adapters, the gating networks of MeteoRA could be fine-tuned easily.   During training, these LoRA adapters remain frozen, and only the gating networks within the Linear modules are trained.
>
> Second, incorporating MoE upcycling into our approach introduces additional challenges:
>
> * **Which LoRA adapter weights should be used as the initialization weights?** Our target scenario involves solving multitask problems with a single model that can autonomously recognize tasks and switch between LoRA adapters. In our experiments, the 28 selected tasks span diverse knowledge domains and task formats. Choosing one LoRA adapter’s weights as the initialization point is not easy given this diversity.
>
> * **Even if an appropriate initialization could be chosen**, training using MoE upcycling would require introducing an auxiliary loss (to achieve load balancing among experts) and keeping all LoRA adapters trainable. This contradicts our objective of directly reusing pre-trained LoRA adapters, as it would introduce additional trainable parameters and computational overhead.
>
> ## W2: Detailed Analysis of Triton Operator
>
> Thank you for your advice. In our submission, we evaluated bmm-torch, bmm-triton, the vanilla MoE forward solution, and single LoRA forward (as a reference) in terms of memory consumption and efficiency. The results were posted in Appendix A.7.
>
> The evaluation focuses on kernel performance (including runtime cost and memory cost) during inference for each forward pass design. One forward pass may involve various factors. The evaluation settings of all factors are listed below:
>
> ```
> batch size (b): [1,16]
> sequence length (s): [64,128,256,512,1024,2048,4096] # present in the x-axis
> top-k selected LoRAs (k): [1,2]
> LoRA rank size (r): [8]
> number of LoRAs (l): [8,28] # 8 LoRAs for classic MoE setting, 28 LoRAs for our setting.
> maximum tokens to generate (g): [32, 256]
> input hidden dimension (h): [4096]
> output hidden dimension (hout): [8192]
> ```
>
> \#combination = b\*k\*l\*g = 2^4=16. Thus, 16 subfigures are shown in Figures 6, 7, and 8, respectively.
>
> As evaluated in Appendix A.7, bmm-torch performs the best in terms of speed, but with large memory consumption for longer sequence lengths (see GB/token in Figure 7). In contrast, bmm-triton achieves a better trade-off between time and space consumption (see Figure 6).
>
> In the main text, we only posted computation consumption due to the page limit. In scaling scenarios to long context (> 8K length) with more LoRA adapters, memory cost could become a bottleneck for bmm-torch. At this point, using bmm-triton to trade 20% speed for significantly reduced memory consumption would become advantageous. We will evaluate the scaling scenario in the future.
>
> ## W3: Figure 3 Legend Size
>
> Thank you for your advice.  We will revise and reorganize the figure to ensure the text does not overlap and increase the font size as much as possible while maintaining clarity.

---

> > ### Comment · Reviewer_dk7r · 2024-11-25
> > **Thank you for the rebuttal**
> >
> > I have read the rebuttal and willing to keep my score.

---

### Official Review · Reviewer_v7tD · 2024-11-03

**Soundness:** 2
**Presentation:** 3
**Contribution:** 2
**Rating:** 5
**Confidence:** 4

**Summary:**

This paper presents MeteoRA, a framework combining MoE and LoRA to enhance inference efficiency via forward acceleration. It analyzes PEFT methods related to user tasks, integrating multiple LoRA adapters for new tokens and identifying the top-k experts for processing. The authors introduce a batched matrix multiplication (bmm-torch) strategy to enable parallel processing of LoRAs, improving speed and efficiency over sequential methods. In summary, by merging MoE and bmm-torch, MeteoRA significantly accelerates token processing and enhances operational efficiency.

**Strengths:**

MeteoRA effectively implements a scalable integration of LoRA while adopting forward acceleration techniques during the inference phase, thereby enhancing the efficiency of the inference process.

**Weaknesses:**

- The paper is not very novel, given that using MoE for LoRA is an idea that has already been extensively explored [1,2,3]. It would be beneficial to clearly delineate how MeteoRA compares to and differs from the referenced LoRAMoE works.

- The term "reuse existing LoRA" is misleading and unclear; it implies the need for offline training and does not introduce any innovation compared to other MoE methods.

- While the bmm-torch method for parallel processing of LoRA adapters improves forward training, it may increase memory consumption. This approach requires larger memory allocations for concurrent processing, potentially offsetting time savings from reduced sequential processing. Please provide quantitative comparisons of memory usage and speed gains across different batch sizes or sequence lengths to clarify this trade-off.


- MeteoRA is presented as an advancement over existing LoRA techniques, a direct comparison with LoRA MoE methods is missing. Such a comparison could underscore the performance superiority of the proposed method. Could the authors conduct and present a detailed comparative analysis with LoRA MoE [1,2,3] methods?

Minior:

-The font size in Figure 3 is too small to read.

References:

[1] When MOE Meets LLMs: Parameter Efficient Fine-tuning for Multi-task Medical Applications, SIGIR 2024.

[2] Mixture of LoRA Experts, ICLR 2024.

[3] Pushing mixture of experts to the limit: Extremely parameter efficient moe for instruction tuning. ICLR 2024.

**Questions:**

The b$\times$s tokens are treated as independent. However, there is concern about potential correlations among tokens. Knowledge across domains can be interrelated, and sentence meaning may depend on context. How do the authors address this issue? Could the assumption of independence negatively affect performance by ignoring relevant interdependencies?

---

> ### Author Response · Authors · 2024-11-22
> **Response to weakness 1 to weakness 3**
>
> ## W1: About Novelty + W2: Misleading Terminology: "Reuse Existing LoRA":
>
> Thank you for your concerns. To begin, we will clarify our motivation and highlight the differences and advantages of our work compared to methods using MoE for LoRA.
>
> **Motivation and Scenario Explanation of MeteoRA**
>
> MeteoRA is designed with the goal of 'integrating numerous existing LoRA adapters into a single base model'. On Huggingface, there are over 25,000 existing LoRA adapters fine-tuned on several popular base LLMs for various specific tasks, and this number continues to grow daily. Our objective is to create a framework that enables a base LLM to 1) incorporate numerous LoRA adapters from LoRA adapter markets, and 2) autonomously select the most appropriate LoRA adapters during inference.
>
> Although we did the offline LoRA fine-tuning for each task, it is only for evaluating our proposed method. In practice, one could apply publicly available LoRA adapters (e.g., downloaded from HuggingFace) to his/her LLM with our MeteoRA (only a few samples per adapter are required). To this end, MeteoRA offers several key advantages:
>
> 1. **Lightweight Setup**: In the MeteoRA module, we employ a Gating network, which is the only component requiring fine-tuning, to integrate LoRA adapters into the base LLM. The trainable parameters across all Gating networks constitute no more than 0.40% of the total parameters in LlaMA3-8B and LlaMA2-13B. Additionally, the data required for fine-tuning can be minimal. In our evaluation, fine-tuning the Gating networks with just 100 samples per task yielded satisfactory performance. We also tested an extreme case by fine-tuning the Gating networks with only 5 samples per task, and MeteoRA still achieved comparable performance.
>
> 2. **Autonomous Task Sensing and Switching**: With its lightweight setup, MeteoRA gains the capability for autonomous task sensing and switching, unlike existing work related to multiple LoRA adapter fusion. Some methods are not designed for reusing LoRA adapters like MeteoRA, while others, such as TIES \[1], DARE \[2] (implemented in Huggingface PEFT), and LoRAHub \[Huang et al., 2023], cannot achieve autonomous task sensing and switching in our 28 tasks evaluation scenario. We will discuss these in detail in the next section.
>
> **Difference between existing LoRA MoE methods and MeteoRA**
>
> The related methods \[Yang et al., 2024 (Moral); Feng et al., 2024 (MoA); Chen et al., 2024 (Llava-mole); Wu et al., 2023d (Mole)] leverage the concept of MoE to construct a MoE-style architecture in LLMs. Most of them require training from scratch for parameters in LoRA modules.
>
> The discussion and detailed evaluation of your mentioned three references are posted in response to W4.
>
> ## W3: Memory and Speed Trade-off
>
> Thank you for highlighting the potential memory concerns with the bmm-torch method.
>
> We have included quantitative comparisons of memory usage and speed gains for various batch sizes and sequence lengths in Appendix A7. In our experiments, all of the LoRA weights account for less than 10% of the base model weights (*e.g. \~7% for 28 lora experts*), resulting in **negligible relative changes** to GPU memory consumption when we apply bmm\_torch for tasks with the regular sequence length.
>
> However, for very long sequence lengths (e.g., 16k+), memory usage may become more significant indeed. To mitigate this, we can **dynamically switch** to the bmm-triton implementation to maintain low memory overhead. Additionally, a **hybrid approach** that combines both bmm-torch and bmm-triton across different layers can be employed to balance their respective advantages.
>
> We hope this addresses your concerns and clarifies the trade-offs involved.

---

> ### Author Response · Authors · 2024-11-22
> **Response to weakness 4, weakness 5, and the question part.**
>
> ## W4: Missing Direct Comparison
>
> Thank you for your suggestions for comparing with LoRA MoE methods. We first discuss the details of each reference:
>
> * **\[1] When MOE Meets LLMs: Parameter Efficient Fine-tuning for Multi-task Medical Applications, SIGIR 2024.**
>
> a. Upon a thorough review of the paper and its accompanying code, we observed that the proposed method requires manual task specification during inference. This limitation prevents it from achieving autonomous task-aware selection and dynamic switching of LoRA adapters based on input, as demonstrated by MeteoRA.
>
> b. Additionally, the study explores a scenario with only 8 tasks, which is relatively limited compared to the broader applicability demonstrated by MeteoRA.
>
> * **\[2] Mixture of LoRA Experts, ICLR 2024. (MoLE)**
>
> a. This paper has not been open-sourced, leaving us to rely solely on its theoretical formulations for implementation.
>
> b. The equations (7) and (8) presented in MoLE paper cannot be used for current mainstream LoRA adapter framework. To make MoLE executable for practical Decoder-only LLM + LoRA adapter paradigm, we choose the gating functions in matrix-wise level posted in MoLE paper.
>
> c. During practical implementation, we found that MoLE imposes significant memory demands, which renders it ineffective for long-text tasks such as CNNDM. This suggests that MoLE may be more suited to computer vision tasks, as highlighted in the original work.
>
> * **\[3] Pushing Mixture of Experts to the Limit: Extremely Parameter Efficient MoE for Instruction Tuning, ICLR 2024. (MoELoRA)**
>
> a. The proposed approach requires all LoRA adapters' fine-tuning in training, which fundamentally differs from the design philosophy of MeteoRA. MeteoRA is designed to directly leverage pre-existing LoRA adapters. This strategy not only reduces the number of parameters involved in training but also isolates adapters for different tasks. For example, when updating a specific LoRA adapter, others remain unaffected, ensuring modularity and task-specific independence.
>
> As discussed above, we did an extended evaluation compared to MoLE and MoELoRA. We implemented MoLE and MoELoRA based on LLaMA3-8B, using the hyperparameters specified in the original paper. These methods were evaluated on 28 tasks we selected, and the results were averaged using the same methodology as in the paper, yielding the following results:
>
> |                | Accuracy    | BLEU  | ROUGE-1 | ROUGE-2 | ROUGE-L |
> | -------------- | ----------- | ----- | ------- | ------- | ------- |
> | MoLE           | 0.343       | 32.38 | 0.135   | 0.037   | 0.098   |
> | MoELoRA        | $\approx 0$ | 2.31  | 0.021   | 0.001   | 0.008   |
> | MeteoRA(T1-1k) | 0.811       | 45.64 | 0.338   | 0.158   | 0.314   |
>
> Two key points regarding the experimental results are worth noting:
>
> 1. Although MoELoRA can produce outputs that adhere to natural language rules, it fails to generate responses in the required format for most tasks and demonstrates poor instruction-following capability. Consequently, its performance on accuracy metrics is nearly zero, and it does not achieve satisfactory results on BLEU or ROUGE scores.
>
> 2. Both MoLE and MoELoRA require significantly higher memory during the inference process, leading to their failure on long-text tasks such as *CNNDM*.
>
> In summary, based on the experimental results, MeteoRA demonstrates substantially better performance.
>
> ## W5: Figure 3 Font Size
>
> Thank you for your advice. We will revise and reorganize the figure to ensure the text does not overlap and increase the font size as much as possible while maintaining clarity.
>
> ## Q1: Token Independence Assumption
>
> Thank you for your insightful question regarding the independence of tokens and potential interdependencies.
>
> In Transformer architectures, all linear transformations—including QKV projections and MLP up/down/gate projections—are inherently token-independent (*each token corresponds to one row vector in the hidden space, and applies right matrix multiplication to project to another space*). The interdependencies among tokens are effectively **captured by the attention mechanism**. Therefore, even the base dense models do not explicitly model interdependencies within the linear transformations themselves, let alone our proposed adapters. As a result, the assumption of token independence in our method does not adversely affect the model's ability to capture relevant interdependencies through the attention mechanism.
>
> We hope this addresses your concern.

---

> > ### Comment · Reviewer_v7tD · 2024-11-25
> >
> > Thanks for the response.  I will raise the score to 5. Major concern is novelty, particularly when compared to existing LoRA MoE and LoRA integrated efforts.

---

### Official Review · Reviewer_oZTQ · 2024-11-04

**Soundness:** 4
**Presentation:** 2
**Contribution:** 3
**Rating:** 8
**Confidence:** 4

**Summary:**

This paper introduces MeteoRA, which automatically applies the appropriate LoRA adapters to a pre-trained LLM based on the current task.
MeteoRA is an MoE-inspired approach in which a gating function selects the top-k LoRA adapters for each token in each input sequence.
The authors demonstrate that MeteoRA performs similarly to an LLM with a handpicked, in-domain LoRA adapter, and provide an efficient
kernel implementation that addresses runtime concerns and memory overhead.

**Strengths:**

1. MeteoRA is a general approach to incorporate domain-specific knowledge from multiple LoRAs in a single model.
2. Extensive evaluation which demonstrates that MeteoRA performs similarly to the PEFT reference implementation, which provides a reasonable upper-bound reference.
3. The authors explain concerns about runtime and memory-efficiency. Based on this, the authors design, implement, and evaluate a CUDA kernel which addresses the concerns.

**Weaknesses:**

1. All LoRAs are stored in GPU memory, which limits the scalability of the approach. In contrast, S-LoRA (a LoRA serving system) scales to thousands of LoRA adapters by swapping LoRA weights to host memory. Proposing a target range for the # of LoRA adapters or a method to swap adapters to host memory could help address this concern.
2. MeteoRA model is fine-tuned on a set of LoRAs and their target domains. Consequently, the approach does not efficiently integrate new LoRA adapters.
3. Capability regression with $k=2$ indicates that LoRA adapters likely interfere with one another. Some discussion of how to mitigate interference or exploit $k=1$ for further speedups could address this weakness.
4. On a few tasks, MeteoRA performs worse than the baselines (e.g., NewsIT, CNNDM, and TrackObj). An explanation of why this might be the case could help contextualize these results.
5. No evaluation on how MeteoRA scales to larger batch sizes. It would be interesting to see the relationship between batch size and runtime/memory because larger batch sizes would access more adapters which could impact these metrics.

**Questions:**

**Questions**
1. How does MeteoRA perform on out-of-distribution tasks (e.g., compared to baselines such as the base LLM, LoRA-F, and LoRA-B)?
2. In section 3.3, what is $p_i$?
3. For measuring forward-pass speed, what is the batch size?

**Suggestions**
1. Introduction should quantify benefit of MeteoRA beyond speedup (e.g., average accuracy increase).
2. Fig 1: unclear where the MoE is located, and how experts are selected.
3. Background: should cite other MoE-based LLMs, such as GLaM (preceded Mixtral), DBRX, and Grok.
4. Section 3.3 needs more revisions for clarity. While I appreciate the explanation to motivate the kernel design, it took several reads to fully understand the problem with the `loop-original` method, why it is 10x slower, and how `bmm-torch` works.
5. Figure 8 is hard to interpret. The font size is small and the colors/lines are difficult to distinguish due to small line width and shading. Such a key figure should be better-presented (i.e., bigger, clearer lines, easier to read).
6. Figure 5: root-of-runtime is a strange (and potentially misnamed) evaluation metric. It would be better to report runtime directly.

---

> ### Author Response · Authors · 2024-11-22
> **Response to weakness 1 and weakness 2**
>
> ## W1: Scalability Limitation
>
> Thank you for your question and suggestion. S-LoRA is designed for LoRA serving, whereas MeteoRA focuses on LoRA adapter reuse with autonomous task sensing and switching. Both methods have unique advantages and characteristics, but S-LoRA requires users to specify the LoRA adapter IDs to be used prior to serving, which is not applicable to our scenario.
>
> In our experiments, all LoRA weights combined account for less than 10% of the base model weights (e.g., \~7% for 28 LoRA experts), resulting in a memory cost of approximately 1.9 GB for LLaMA3-8B equipped with MeteoRA in BF16 format. For instance, loading 100 LoRA adapters simultaneously would require less than 8 GB of memory. Additionally, since MeteoRA employs an MoE-style gating mechanism, the memory cost associated with activations remains constant, regardless of the number of loaded LoRA adapters.
>
> The concept of S-LoRA is orthogonal to MeteoRA, and our future research will explore adapter off-loading strategies suitable for MeteoRA by incorporating insights from S-LoRA or other related works. Thanks for your advice again.
>
> ## W2: Inefficient Integration of New LoRA Adapters
>
> Thank you for your question. In our paper, we included three baseline methods that do not require re-training when new LoRA adapters are integrated: TIES, DARE, and Arrow, with the first two supported by HuggingFace's PEFT library. Here is a brief comparison of these methods:
>
> * **TIES and DARE**: These methods aim to merge models fine-tuned for different downstream tasks. While HuggingFace’s PEFT library provides support for these strategies, their performance significantly deteriorates when merging LoRA adapters across 28 tasks. Furthermore, these methods require manual adjustment of the weights for each LoRA adapter, meaning they cannot automatically switch to the appropriate LoRA based on user input.
>
> * **Arrow**: This method uses a routing algorithm to reuse LoRA adapters, leveraging the right first singular vector derived from the SVD of the product $A_iB_i^T$ for each LoRA adapter.
>
> As shown in Table 1 of our paper, TIES and DARE exhibit significant performance drops if the weights of each LoRA adapter are not carefully tuned. Similarly, Arrow struggles to achieve satisfactory performance on the scale of 28 tasks.
>
> From our point of view, learning a model once to merge an arbitrary set of LoRA adapters is not applicable. Given an arbitrary set of LoRA adapters (sampled from a universal set of LoRA adapters), each LoRA adapter is tuned on a task-specific distribution. Assume that a learner, A, can learn the optimal weight for LoRA Merge. Since different tasks have different data distributions, the corresponding LoRA adapters are fitted to different distributions. If the optimal weight in learner A exists, learner A would be a "universal learner." However, the No Free Lunch Theorem tells us that no such learner can exist. This means that learner A cannot learn a set of weights to merge all different LoRA adapters to perform well across all inference tasks.

---

> > ### Author Response · Authors · 2024-11-22
> > **Response to weakness 3 to weakness 5, Questions**
> >
> > ## W3 & W4: Capability Regression
> >
> > Thank you for your question. We would like to provide further clarification on these two issues. In MeteoRA, the performance drop on certain tasks when $k=2$ or even $k=1$ arises from a combination of factors. Based on our experimental setup, the following influences are noteworthy:
> >
> > 1. **Base Model Capability**: The overall performance of MeteoRA on LLaMA3-8B is slightly better than on LLaMA2-13B, indicating that the base model's capacity affects the results.
> >
> > 2. **Task Selection**: When selecting tasks, we prioritized diversity in task formats and domains while intentionally including some tasks with overlapping characteristics to observe potential interactions. With 28 tasks, conflicts between the corresponding LoRA adapters are inevitable, which can lead to performance degradation.
> >
> > 3. **LoRA Adapter Quality**: Since the output of one LoRA adapter affects the subsequent Gating Network’s choices, the quality of individual LoRA adapters can impact MeteoRA’s overall performance.
> >
> > 4. **Training Approach**: The LoRA-F baseline combines the full training datasets of all tasks into a single concatenated dataset. This method inherently prioritizes tasks with larger datasets (e.g., CNNDM) while significantly compromising performance on tasks with smaller datasets (e.g., ObjCount). In contrast, our approach fine-tuned MeteoRA using a maximum of 1,000 samples per task. Consequently, while MeteoRA may exhibit slight underperformance on certain tasks that require larger datasets to fine-tune the gating mechanism effectively, this trade-off aligns with our focus on balanced and efficient task performance.
> >
> > Additionally, MeteoRA's primary trade-off is ensuring that all tasks maintain reasonably consistent performance without significant degradation. While performance on a small subset of tasks may slightly lag behind the baseline, the overall gap between MeteoRA and PEFT is minimal, and both methods achieve satisfactory results across tasks.
> >
> > ## W5: Scaling with Batch Size
> >
> > Thank you for your question. In MeteoRA, the behavior of batch size during model inference is identical to that in the standard HuggingFace Transformers library. During evaluation, we typically set the batch size to 16 or 8. Samples within the same batch are independent of each other, consistent with standard model inference processes.
> >
> > ## Q1: Out-of-Distribution Performance
> >
> > Thank you for your question. For this evaluation, we selected three popular datasets from HuggingFace:
> >
> > * ai2\_arc: https://huggingface.co/datasets/allenai/ai2_arc
> >
> > * BoolQ: https://huggingface.co/datasets/google/boolq
> >
> > * databricks\_dolly\_15k: https://huggingface.co/datasets/databricks/databricks-dolly-15k
> >
> > We evaluated MeteoRA on these datasets using LLaMA 3-8B. The results demonstrate that MeteoRA can be considered a model fine-tuned on instruction-following datasets. It is able to provide appropriate responses based on the instructions in the tasks and correctly terminate responses with `<eos>`, rather than continuing to generate text indefinitely as the base model might. Furthermore, there is no significant performance difference between MeteoRA and LoRA-B or LoRA-F.
> >
> > ## Q2: Clarification on $p_i$
> >
> > Thank you very much for your feedback. Regarding your question about $p_i$ in Section 3.3, it denotes the number of tokens assigned to the $i$-th LoRA expert. We have provided a detailed explanation of this parameter on line 214 of the manuscript for your reference.
> >
> > ## Q3: Batch Size for Forward-Pass Speed Measurement
> >
> > Thank you for your question. For measuring forward-pass speed, the batch size refers to the first dimension in the standard "bshd" or "bsh" input layout. It is used to compare the efficiency impact of processing different numbers of independent sequences simultaneously.

---

> ### Author Response · Authors · 2024-11-22
> **Response to suggestions**
>
> ## S1: Lack of Quantified Benefits Beyond Speedup
>
> Thank you for your suggestion. We will revise the Introduction to include additional quantitative results.
>
> ## S2: Unclear MoE Details in Figure 1
>
> Thank you for pointing this out. We will revise Figure 1 to make these details more explicit and ensure the information is clearly presented.
>
> ## S3: Missing Citations for MoE-based LLMs
>
> Thank you for pointing this out. We will revise the paper to include the relevant citations in the next version.
>
> ## S4: Clarity Issues in Section 3.3
>
> Thank you for your valuable suggestion regarding Section 3.3.
>
> * **Loop-original**: This method utilizes a for-loop paradigm, where in each iteration, only the tokens assigned to the $i$-th LoRA expert are processed. Consequently, every loop must traverse all LoRA experts, making it a LoRA *expert-centric* approach. When the token assignment is so sparse that each iteration handles few tokens, the efficiency drops significantly, resulting in the observed 10x slowdown when the number of experts rises to 28+, as well as the increase of the sequence length.
>
> * **Bmm-torch**: In contrast, this approach is *token-centric*. It assigns LoRA experts to tokens, allowing each token to independently perform LoRA operations (two matrix multiplications). By leveraging PyTorch's `bmm` function, all tokens can execute these operations in parallel using two batch matrix multiplications. This parallel processing substantially improves efficiency compared to the loop-original method above.
>
> We will revise Section 3.3 in the next version.
>
> ## S5: Presentation Issues in Figure 8
>
> Thank you for your advice. We will revise and reorganize the figure to ensure the text does not overlap and increase the font size as much as possible while maintaining clarity.
>
> ## S6: Potential Misnaming of Metric in Figure 5
>
> Thank you for your suggestion. Initially, we attempted to use runtime as the metric. However, since a smaller runtime indicates higher efficiency, the fastest baseline, *single-lora*, would appear very close to zero, and other bins would overlap with the baseline or become compressed, making the chart difficult to interpret. To address this, we used the reciprocal of runtime as the metric so that the y-axis represents "speed." This allows the fastest baseline to appear at the top of the chart, with the other bins more clearly differentiated.

---

### Author Response · Authors · 2024-11-27

We sincerely appreciate the valuable suggestions provided by all the reviewers. We have revised the paper based on the suggestions and have submitted an updated version. We would be grateful to address any further questions or engage in any additional discussions, if needed.

---

> ### Author Response · Authors · 2024-12-03
>
> Dear Reviewers,
>
> Thank you once again for reviewing our paper and providing valuable feedback. As the deadline approaches, we would like to check if our rebuttal has adequately addressed your concerns. Should you have any further questions or require clarification, please feel free to let us know. We are more than happy to address any additional inquiries.
>
> MeteoRA authors

---

### Meta-Review · Area_Chair_wkH6 · 2024-12-20

**Metareview:**

The paper proposes a framework to incorporate domain-specific knowledge from multiple LoRAs in a single model, integrating multiple LoRA adapters for new tokens and identifying the top-k experts for processing.  A batched matrix multiplication (bmm-torch) strategy accelerates token processing. An evaluation using the LlaMA2-13B and LlaMA3-8B base models equipped with 28 existing LoRA adapters is shown to be performant compared to fine tuning, particularly in handling composite tasks.

Reviewers found the work sufficiently different from prior art, and were impressed by the set of experiments, all the way to an implementation of a CUDA kernel. Issues regarding scaling to multiple GPUs were brought up, and are left as future work. One reviewer remained concerned regarding novelty, but others were not as concerned.

**Additional Comments On Reviewer Discussion:**

Reviewers appreciated the additional out-of-distribution experiments, the multitude of smaller/presentation issues addressed. The comparisons to additional competitor methods (some experimental) should be added to the paper.

One reviewer remained concerned that "root-of-runtime" is not an appropriate metric, especially since it is used solely for data point visibility. Inference runtime or throughput (batches/sec) would be more appropriate. The reviewer also correctly suggests that to resolve visibility issues by using logarithmic scaling: this is indeed more scholarly/consistent with best practice.

Notation should be made uniform between the main body and the appendix.

---

### Decision · Program_Chairs · 2025-01-22

Accept (Poster)